# Mixture of In-Context Prompters for Tabular PFNs

**Derek Xu**
University of California, Los Angeles
derekqxu@ucla.edu

**Olcay Cirit**
Uber Technologies Inc

**Reza Asadi**
Uber Technologies Inc

**Yizhou Sun**
University of California, Los Angeles

**Wei Wang**
University of California, Los Angeles

## Abstract

Recent benchmarks find In-Context Learning (ICL) outperforms both deep learning and tree-based algorithms on small tabular datasets. However, on larger datasets, ICL for tabular learning suffers in both efficiency and effectiveness. In terms of efficiency, transformers incur linear space and quadratic time complexity w.r.t. context size. In terms of effectiveness, contexts at inference encounter distribution shift compared to contexts from pretraining. We propose MixturePFN, which extends Sparse Mixture of Experts to the state-of-the-art ICL for tabular learning model. Specifically, MixturePFN finetunes a specialized ICL expert on each cluster of tabular data and routes new test samples to appropriate experts at inference. MixturePFN supports constant-size contexts by splitting large training datasets into more manageable clusters. MixturePFN addresses distribution shift by finetuning an expert on each training dataset cluster via bootstrapping. Extensive experimental results shows MixturePFN outperforms 19 baselines both in mean rank and as the Condorcet winner across 36 diverse tabular datasets.

## 1 Introduction

Tabular data is a popular data format across various domains, consisting of column-wise features and row-wise data samples. Each feature can be either continuous, categorical, or ordinal. Thanks to the prevalence of relational databases, which ensure data integrity, consistency, and low redundancy, tabular data is widely used across various domains such as medicine, finance, and advertising. Hence, improving learning algorithms on tabular data is of interest to many researchers.

General tabular datasets remain unconquered by most deep learning algorithms (Popov et al., 2019; Gorishniy et al., 2021; Somepalli et al., 2021; Arik & Pfister, 2021; Yamada et al., 2020; Yoon et al., 2020; Chen et al., 2022). Instead, gradient-boosted decision trees (GBDTs) (Chen & Guestrin, 2016; Prokhorenkova et al., 2018), achieve better overall performance on tabular benchmarks (McElfresh et al., 2023; Shwartz-Ziv & Armon, 2022) considering a wide range of number of samples, numbers of features, feature types, and feature distributions. Recently, transformer-based prior-fitted networks, PFNs (Hollmann et al., 2022), have garnered interest, for their surprisingly strong and state-of-the-art performance on small tabular datasets with $\leq 3,000$ samples (Hollmann et al., 2022; McElfresh et al., 2023).

Unlike SGD-based deep learning, which trains a model on each downstream dataset, PFNs learn the underlying patterns behind how downstream training datasets are generated. Specifically, PFNs are first pretrained on datasets generated from a handcrafted dataset prior (Müller et al., 2021; Von Oswald et al., 2023). Then, during inference, PFNs predicts test samples by modeling the underlying generative process of the observed training dataset. In this sense, PFNs are a form of In-Context Learning (ICL) (Brown et al., 2020), where the context consists of downstream training samples. In practice, PFNs are modeled as a transformer, where each dataset sample is converted into a token and each dataset is converted into a prompt. By pretraining on multiple datasets generated

by the handcrafted prior, PFN learns a better inductive bias than both conventional deep learning algorithms and gradient-boosted decision trees.

One reason PFN performance drops on larger datasets is due to how it handles efficiency bottlenecks. First, PFN inference is computationally expensive. Because the entire dataset is fed to the transformer as a "prompt", the inference time and space complexity for $N_{train}$ training samples are $\mathcal{O}(N_{train})$ (Dao et al., 2022; Dao, 2023) and $\mathcal{O}(N_{train}^2)$ respectively. This is in stark contrast to GBDTs and traditional deep learning approaches where the inference time and space complexity is $\mathcal{O}(1)$. To keep inference time low, existing works (Hollmann et al., 2022; McElfresh et al., 2023) randomly sample from the training dataset to form the "prompt." However, randomly sampling the training dataset could throw away useful training samples, leading to information loss in the context.

Another reason PFN performance drops could be distribution shift between the inference-time context dataset and the pretraining dataset generating prior. Because the prior is handcrafted, large real-world datasets could follow a different underlying dataset generating process than the one seen during pretraining. If this is the case, additional finetuning computation is required to align the pretraining and inference-time dataset generating priors. We justify such computation by comparing against baselines that also perform finetuning on downstream data.

In this work, we analyze recent claims (McElfresh et al., 2023) on PFN's effectiveness, finding PFN performance does not scale w.r.t. dataset size. We improve PFN's performance w.r.t. dataset size[1] by proposing Sparse "Mixture of In-Context Prompters" (MICP), which creates scalable "prompts" by routing new test samples to a prompter specialized context relevant to said test sample. We solve PFN's distribution shift limitations with "Context-Aware PFN" (CAPFN), which finetunes PFNs for downstream datasets via bootstrapping. We call our combined model MIXTUREPFN. To summarize:

- To improve scalability, we are the first to propose Sparse Mixture of In-Context Prompters (MICP) which routes new test samples to a pool of scalable prompters for In-Context Learning. MICP efficiently routes informative context to downstream testing points with $\mathcal{O}(1)$ memory and $\mathcal{O}(log(N_{train}))$ time complexity, w.r.t. training dataset size, $N_{train}$.

- To improve performance, we finetune Context-Aware Prior-Fitted Network (CAPFN), which aligns pretrained PFNs with inference-time data using a novel bootstrapping policy.

- MIXTUREPFN scales transformer PFNs to tabular datasets with much larger number of samples, encountering minimal performance deterioration w.r.t. dataset size.

- MIXTUREPFN both achieves the highest mean rank with statistical significance and is the Condorcet winner across 36 diverse tabular datasets against 19 strong deep learning and tree-based baselines. We will release our code on Github.

## 2 PRELIMINARIES

We consider tabular classification problems, where the inputs are numerical, ordinal, or categorical columns encoded as a $d$-dimensional feature vector, $x \in \mathbb{R}^d$, the output is the corresponding label, $y \in [1, ..., C]$, and the dataset consists of labelled input output pairs, $D = \{(x^{(i)}, y^{(i)})\}_{i=0}^N$.[2] Given the training dataset, $D_{train} = \{(x_{train}^{(i)}, y_{train}^{(i)})\}_{i=0}^{N_{train}}$, and test samples, $X_{test} = [x_{test}^{(i)}]_{i=0}^{N_{test}}$, our goal is to correctly predict the corresponding test labels, $Y_{test} = [y_{test}^{(i)}]_{i=0}^{N_{test}}$. MIXTUREPFN is inspired by Prior Fitted Networks, which we first introduce.

### 2.1 PRIOR FITTED NETWORKS

Prior Fitted Network (PFN) (Müller et al., 2021) is a parameterized model, $q_\theta$, with weights, $\theta$, that learns to approximate Bayesian inference given the dataset prior, $p(D)$, via In-Context Learning (ICL) (Brown et al., 2020). Specifically, PFN inference approximates the posterior predictive

---

[1]We acknowledge PFN also encounter scalability challenges in terms of feature count and label size, but we choose to tackle dataset size first, as it is the most apparent scalability bottleneck.

[2]We provide a table with all math notations in the Appendix.

distribution (PPD), $p_\theta(y|x, D) = \int_\phi p(y|x, \phi)p(D|\phi)p(\phi)d\phi$, where $\phi$ is the hypothesis mechanism behind how the tabular data is generated. For example, $\phi$ can be a structural causal model. Refer to Bayesian inference transformers (Müller et al., 2021) for further details.

### 2.1.1 PRETRAINING

To approximate the PPD, PFNs are pretrained to minimize KL-Divergence between the parameterized model, $q_\theta(y|x, D)$, and the PPD, $p(y|x, D)$, over the dataset prior, $p(D)$, which was proven equivalent to optimizing the prior data negative log likelihood, $\mathcal{L}_{\text{PFN}}$. As shown in Equation 1 (Müller et al., 2021), this loss iteratively samples new datasets from a handcrafted dataset prior, $p(D)$, via Monte-Carlo.

$$\mathcal{L}_{\text{PFN}} = \mathop{\mathbb{E}}_{x,y,D\sim p(D)} [-log(q_\theta(y|x, D))] \tag{1}$$

TABPFN (Hollmann et al., 2022) is the state-of-the-art pretrained PFN transformer for tabular data. It treats the hypotheses, $\phi$, as randomly sampled structural causal models (SCM) (Pearl, 2009; Peters et al., 2017) mixed with the original Bayesian Neural Network prior (Müller et al., 2021). Training dataset samples are generated by first sampling a SCM graph, $\phi \sim p(\phi)$, followed by sampling the SCM, $x, y, D \sim p(D|\phi)$.

Transformer-based (Vaswani et al., 2017) PFNs tokenize the sampled dataset, $(x, D)$ as input to the parameterized model, $q_\theta$, as shown in Figure 6 and discussed in Section 2.1.2. Note, PFN inputs are analogous to "prompts" from In-Context Learning (ICL) (Brown et al., 2020; Dong et al., 2022; Xu et al., 2024), hence they are called "prompts" in this work.

### 2.1.2 INFERENCE

During inference, transformer-based PFNs tokenize the downstream dataset, $(X_{test}, D_{train})$, into batched "prompts", consisting of $N_{train}$ encoder tokens and $N_{batch}$ decoder tokens, where each data sample corresponds with one token.[3] Because tabular columns are permutation invariant, TABPFN shuffles feature orderings and scalings, running $q_\theta$ on each permutation of the "prompt", then returning an ensembled prediction. We provide illustrations of this process in the Appendix. TABPFN does not perform finetuning, only inference, on downstream datasets.

### 2.1.3 EFFICIENCY LIMITS EFFECTIVENESS

Full-prompt TABPFN cannot scale to datasets with large numbers of training samples (McElfresh et al., 2023). Because the prompt size is directly correlated with the training dataset size, $N_{train}$, encoding the entire training dataset into TABPFN's prompt incurs significant compute costs. Hence, when the training dataset is too large, **existing works randomly sample $B$ out of $N_{train}$ train samples to form the context, leading to information loss.** We call this prompting approach TABPFN*. Most existing works (Hollmann et al., 2022; McElfresh et al., 2023) set $B = 3000$.

TABPFN has an important distinction in its batching protocol. Natively, TABPFN forward pass is less efficient than Large Language Models (LLMs) because each prompt is duplicated across $N_{ensemble}$ feature shufflings and scalings to ensure table column invariance. To counteract this, TABPFN forms more efficient prompts, fitting $N_{batch}$ test samples and the context in the same prompt. In contrast, In-Context Learning (ICL) with LLMs (Liu et al., 2021) assigns a unique prompt and context to each test sample. Hence, TABPFN balances its $N_{ensemble}\times$ slower inference by storing $N_{batch}\times$ more efficient prompts. **As a consequence, PFN prompting strategies must support PFN-style batching, where multiple test points share the same context, to perform efficient inference.**

Finally, existing works (Hollmann et al., 2022; McElfresh et al., 2023) utilize TABPFN for zero-shot inference. While this brings obvious efficiency benefits by removing reliance on downstream training data, **TABPFN assumes minimal distribution shift between pretraining and inference.** Furthermore, because TABPFN is trained on datasets sampled from a dataset prior, not training samples sampled from a pretraining dataset, finetuning TABPFN is nontrivial.

---

[3]Our dataset is split into train/dev/test sets. During hyperparameter tuning, decoder tokens are taken from the dev set instead.

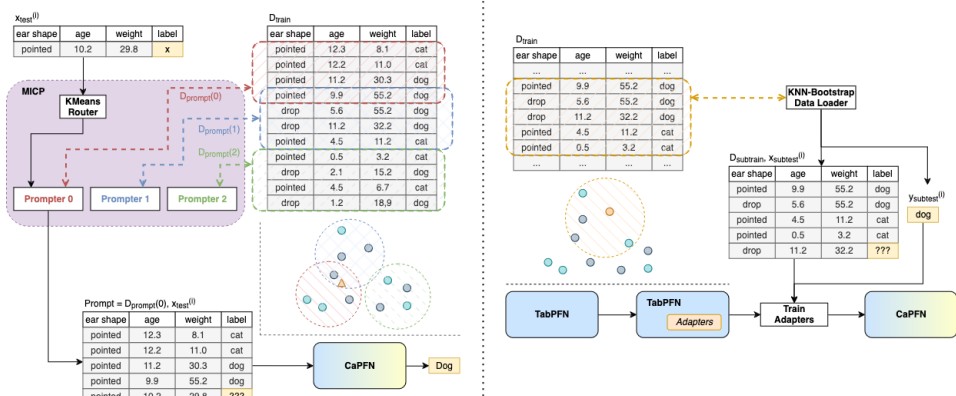

Figure 1: Illustration of MIXTUREPFN. **MICP (Left)**: New test samples are passed to a router that picks 1 out of $K$ prompters to form a scalable "prompt" with $B$ training samples for the downstream PFN model. **CAPFN (Right)**: TABPFN is frozen, fitted with adapters, then finetuned using data prior negative loss likelihood, Equation 1, on our bootstrapped data prior, $p(D|D_{train})$. This prior simulates the MICP inference mechanism. The finetuned model is called CAPFN.

## 3  METHOD

To improve both the efficiency and effectiveness of TABPFN (Hollmann et al., 2022), we propose a method to combine Sparse Mixture of Experts (Shazeer et al., 2017; Lewis et al., 2021) with Prior-Fitted Networks (Müller et al., 2021): MIXTUREPFN.

MIXTUREPFN is composed of 2 parts: **Mixture of In-Context Prompts (MICP)**, which define how to seperate large training datasets into manageable clusters, and **Context-Aware Finetuning (CAPFN)**, which finetunes an expert PFN model on each specialized cluster. In terms of efficiency, MICP is an efficient inference-time prompting technique. Thus, MICP, in isolation, performs zero-shot inference like TABPFN (Hollmann et al., 2022).

Under MICP, we hypothesize distribution shift exists between pretraining and inference. To address said distribution shift, we use bootstrapping to finetune each MICP cluster's transformer model. We find this additional finetuning cost is necessary to close the gap between TABPFN (Hollmann et al., 2022) and other finetuning algorithms, like XGBOOST (Chen & Guestrin, 2016).

### 3.1  MIXTURE OF IN-CONTEXT PROMPTERS (MICP)

The goal of MICP is to construct "prompts" that retains more information than TABPFN*'s random sampling policy. Inspired by Sparse Mixture of Experts (Shazeer et al., 2017; Lewis et al., 2021), where each test sample is routed to an specialized expert trained on a subset of the training dataset, Sparse Mixture of In-Context Prompters, MICP, routes each test sample to one of $K$ "In-Context Prompters" (ICP), $\{\mathcal{T}_k\}_{k=0}^K$, specializing on a cluster of the training dataset, using a routing module, $\mathcal{R} : \mathbb{R}^d \rightarrow \{0, ..., K-1\}$. Each ICP then constructs a relevant "prompt' for incoming test samples, which are sent to the downstream PFN model in batch. Specifically, ICP concatenates incoming test samples, $\{x_{test}^{(i)} : \mathcal{R}(x_{test}^{(i)} = k)\}$, with its training cluster context, $D_{prompt}(k) \subset D_{train}$, to form the scalable "prompt": $\mathcal{T}_k(\{x_{test}^{(i)} : \mathcal{R}(x_{test}^{(i)} = k)\}) = (\{x_{test}^{(i)} : \mathcal{R}(x_{test}^{(i)} = k)\}, D_{prompt}(k))$.

Because larger clusters improve PFN performance (Hollmann et al., 2022; McElfresh et al., 2023) at the cost of efficiency (as we show in Section 5.3), existing works (McElfresh et al., 2023) adopt a fixed prompt size at $B = 3000$. For fair comparison, we also follow this setting, where each prompter's context collects a training cluster context of fixed size, $|D_{prompt}(k)| = B$.

Compared to TABPFN*, which construct random global context for each test point, **MICP**constructs prompts with relevant local context for each test point. Hence, **MICP retains more useful information than TABPFN*.** To construct local clusters, we run K-Means on the training dataset: $\{D_{cluster}^{(k)}\}_{k=0}^K, \{x_{center}^{(k)}\}_{k=0}^K = \text{KMEANS}(D_{train})$. To construct contexts of a

| Method | Condorcet Statistics | | | | All Algo. | Top-10 Algo. |
| | #Votes↑ | #Wins↑ | #Ties | #Losses↓ | Mean ± Std Rank↓ | |
|---|---|---|---|---|---|---|
| MixturePFN | **524** | **19** | 0 | **0** | **2.350 ± 1.824** | **2.273 ± 1.7106** |
| XGBoost | 500 | 18 | 0 | 1 | 5.500 ± 4.621 | 4.000 ± 2.663 |
| CatBoost | 474 | 17 | 0 | 2 | 4.900 ± 4.158 | 3.955 ± 2.688 |
| SAINT | 408 | 16 | 0 | 3 | 8.300 ± 5.367 | 4.045 ± 1.965 |
| TabPFN* | 381 | 13 | 1 | 5 | 4.550 ± 2.747 | 4.040 ± 1.311 |
| LightGBM | 373 | 14 | 1 | 4 | 9.150 ± 4.351 | 6.409 ± 2.839 |
| DANet | 312 | 14 | 0 | 5 | 9.050 ± 3.369 | 7.045 ± 1.988 |
| FTTransformer | 294 | 12 | 0 | 7 | 8.600 ± 3.541 | 6.773 ± 2.235 |
| ResNet | 286 | 11 | 0 | 8 | 8.400 ± 3.262 | 6.864 ± 1.961 |
| SVM | 285 | 9 | 0 | 10 | 11.300 ± 4.766 | 7.500 ± 2.482 |
| STG | 284 | 10 | 0 | 9 | 11.900 ± 4.549 | - |
| RandomForest | 247 | 7 | 0 | 12 | 11.600 ± 4.443 | - |
| NODE | 243 | 7 | 0 | 12 | 13.350 ± 3.410 | - |
| MLP-rtdl | 227 | 5 | 0 | 14 | 10.800 ± 5.046 | - |
| TabNet | 210 | 5 | 0 | 14 | 13.550 ± 5.296 | - |
| LinearModel | 202 | 3 | 1 | 15 | 12.400 ± 4.652 | - |
| MLP | 191 | 5 | 1 | 13 | 13.700 ± 3.621 | - |
| VIME | 134 | 2 | 0 | 17 | 15.350 ± 3.851 | - |
| DecisionTree | 114 | 1 | 0 | 18 | 16.800 ± 3.881 | - |
| KNN | 74 | 0 | 0 | 19 | 18.450 ± 1.936 | - |

Table 1: MIXTUREPFN is the Condorcet winner across 36 datasets against 19 baseline algorithms. MIXTUREPFN achieves the top mean rank across 20 datasets where all algorithms successfully run and across 22 datasets where all Top-10 algorithms successfully run. To break ties, we rank algorithms based on their mean log-likelihoods following TABZILLA (McElfresh et al., 2023). We report the Condorcet matrix, dataset breakdowns, and accuracy-metric results in the Appendix.

desired size, we expand K-Means clusters with less than $B$ samples, $\text{KNN}(x_{center}^{(k)}|D_{train}, B)$, and subsample K-Means clusters with more than $B$ samples, $\text{SAMPLE}(D_{cluster}^{(k)}, B)$ to form the context, $D_{prompt}(k)$. Our router assigns test points to the nearest cluster center: $\mathcal{R}(x) = \text{NNS}(x|\{x_{center}^{(k)}\}_{k=0}^{K})$. **MICP supports PFN-style batching** by grouping test points that belong to the same ICP cluster: $\mathcal{R}(x_{batch}^{(i)}) = \mathcal{R}(x_{batch}^{(j)}) \forall i, j$ into a batched test "prompts": $(X_{batch}, D_{prompt}^{(k)})$, where multiple test points, $X_{batch} \subseteq X_{test}$, share the same context.

In total, router and prompter initialization takes $\mathcal{O}(tN_{train}K + (N_{train} + KB)logN_{train})$ time and $\mathcal{O}(N_{train} + KB)$ space complexity and is done once before inference. Routing takes $\mathcal{O}(log(K))$ time and $\mathcal{O}(1)$ space complexity, using efficient nearest neighbor search with ball-tree for each test sample. PFN transformer inference takes $\mathcal{O}(B^2 + BN_{batch})$ time and space complexity, as MICP prompts contain at most $B$ training samples and $N_{batch} = |X_{batch}|$ testing samples. We provide time and space complexity details in the Appendix. We illustrate MICP in Figure 1.

### 3.1.1 EFFICIENCY AND EFFECTIVENESS TRADE-OFF

The effectiveness of MICP prompts depend on the number of ICPs used, $K$. As the complexity and size of data increase, more ICPs are needed to capture the entropy of the labels. This is natural as each router's support set, $D_{prompt}(\mathcal{R}(x_{test}^{(i)}))$, should be representative of test samples routed to that cluster, $\text{KNN}(x_{test}^{(i)}|D_{train}, B)$, which we call the support set. If the true support set becomes more granular as the dataset size increases, more ICPs are required to maximize overlap: $|D_{prompt}(\mathcal{R}(x_{test}^{(i)})) \cap \text{KNN}(x_{test}^{(i)}|D_{train}, B)|$.

We theoretically characterize this relationship between $K$, $B$, and overlap by analyzing conditions required for nonzero overlap on the training data: $|D_{prompt}(\mathcal{R}(x_{train}^{(i)})) \cap \text{KNN}(x_{train}^{(i)}|D_{train}, B)| \geq 1 \, \forall i \in [0, ..., N_{train} - 1]$. Specifically, we encourage nonzero overlap

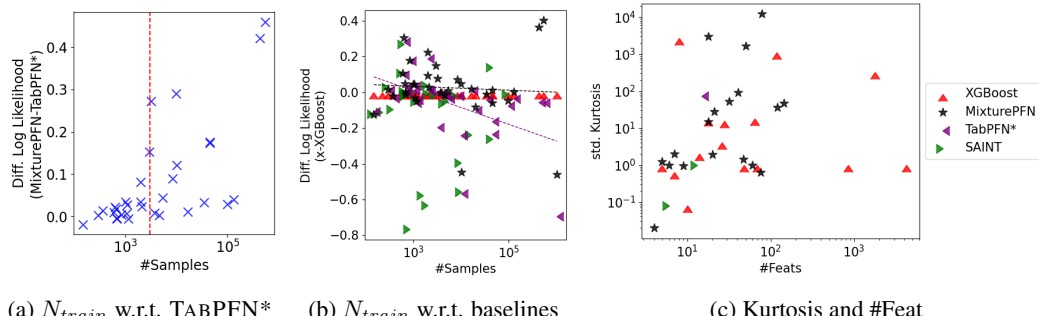

(a) $N_{train}$ w.r.t. TABPFN*    (b) $N_{train}$ w.r.t. baselines    (c) Kurtosis and #Feat

Figure 2: **(a):** We plot the difference in Log Likelihood between MIXTUREPFN and TABPFN* for each dataset of size $N_{train}$. MIXTUREPFN substantially improves the performance and TABPFN* and runs on datasets with $> 3,000$ samples. **(b):** We plot the Log Likelihood of the top deep learning (DL) PFN, and tree baselines across all 36 datasets and the best-fit line between rank and dataset size, compared to the top baseline. Unlike TABPFN, MIXTUREPFN maintains its good performance as the size of the dataset increases. **(c) :** We plot the best among the top DL, PFN, and tree baselines on all 36 datasets across different dataset properties. MIXTUREPFN performs well across different dataset irregularities. We provide further breakdowns in the Appendix.

by scaling the number of "prompts", $K$, linearly with the size of each "prompt", $B$, and training dataset size, $|N_{train}|$, as stated in Theorem 3.1.1: $K \geq \lceil N_{train}/B \rceil$.

[Nonzero Overlap] If every K-Means cluster contains at most $B$ samples, $|D_{cluster}^{(k)}| \leq B \ \forall k \in [0, ..., K-1]$ and training points route to their assigned K-Means cluster $\mathcal{R}^*(x_{train}^{(i)}) = k : x_{train}^{(i)} \in D_{cluster}^{(k)}$ [4], then nonzero overlap on the training data is guaranteed, $|D_{prompt}(\mathcal{R}^*(x_{train}^{(i)})) \cap \text{KNN}(x_{train}^{(i)}|D_{train}, B)| \geq 1 \ \forall i \in [0, ..., N_{train}-1] \ \forall D_{train}$.

This insight allows MIXTUREPFN to trade-off efficiency and effectiveness with a single hyperparameter, $\gamma$, which controls the number of ICPs as a ratio of training and support set sizes: $K = \lceil \gamma N_{train}/B \rceil$. Intuitively, larger $\gamma$ improves effectiveness at the cost of efficiency. **Assuming fixed $\gamma$, $N_{batch}$, and $B$, MIXTUREPFN routing takes $\mathcal{O}(log(N_{train}))$ time and $\mathcal{O}(1)$ space complexity, and PFN inference takes $\mathcal{O}(1)$ time and space complexity.**

## 3.2 CONTEXT-AWARE FINETUNING (CAPFN)

The goal of CAPFN is to counteract distribution shift by specializing each prompter on its assigned context via parameter efficient finetuning. PFNs are pretrained on the ICL task over a synthetic dataset prior, $p(D) = p(D|\phi)p(\phi)$ (Müller et al., 2021; Hollmann et al., 2022). Inspired by recent works which aligns Large Language Models on ICL "prompts" via finetuning (Thoppilan et al., 2022; Wei et al., 2021; Gu et al., 2023), we argue the pretraining data prior, $p(D) = p(D|\phi)p(\phi)$, is different than the true data generating mechanism during inference, $p(D_{prompt}|D_{train})$, which was described in Section 3.1. This leads to distribution shift. To better align the parameterized model, $q_\theta$, with the inference-time dataset, $D_{prompt}$, CAPFN uses bootstrapping on the downstream dataset, $D_{train}$, to simulate ICL "prompts": $(X_{subtest}, Y_{subtest}, D_{subtrain}) \sim p(D|D_{train})$, where $X_{subtest} \subset X_{train}$, $Y_{subtest} \subset Y_{train}$, and $D_{subtrain} \subset D_{train}$. Bootstrapped samples are used to tune adapters (Houlsby et al., 2019) via prior data negative log likelihood loss, as shown in Equation 1, except the dataset prior is now the bootstrap mechanism: $p(D) = p(D|D_{train})$.

---

[4]Thse conditions can be satisfied via constrained K-Means (Bradley et al., 2000), which ensures each cluster has at most $B$ entries, and a router that sends train points to their assigned clusters. In practice, we find the relationship with the tunable parameter $\gamma$ also holds for MIXTUREPFN's regular K-Means and Nearest-Neighbor Search router.

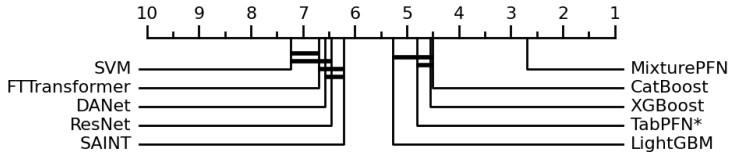

Figure 3: Wilcoxon-Signed Rank Test shows MIXTUREPFN significantly outperforms the Top-10 baselines on the 22 shared datasets. To break ties, we rank algorithms based on their mean log-likelihoods following TABZILLA (McElfresh et al., 2023). We compute the rank across all 10 cross-validation splits. We report additional critical difference diagrams in the Appendix.

### 3.2.1 BOOTSTRAPPING LARGE MICP DATASETS

The bootstrap procedure mimics MICP on large $N_{train} > 3000$ datasets: $p(D|D_{train}) = p(D_{support}|x)p(x|D_{train})$. Specifically, we sample a random training point from the training dataset, $x \sim p(x|D_{train})$, then run K-Nearest Neighbors from the sampled point, $p(D_{support}|x) = \text{KNN}(x|D_{train}, B)$, as defined in Section 2.1.3, to obtain a bootstrap dataset, $D_{bootstrap}$. We randomly split the bootstrapped dataset $D_{bootstrap} \sim p(D|D_{train})$ into train/test splits to obtain the "labelled prompt", $(X_{subtest}, Y_{subtest}, D_{subtrain})$.

### 3.2.2 BOOTSTRAPPING SMALL DATASETS

MICP does not run on smaller $N_{train} \leq 3000$ datasets. However, bootstrapping can still be used to finetune the model on said datasets to match the downstream dataset distribution. In this case, we sample from $p(D|D_{train})$ by randomly sampling 90% of training samples without replacement to obtain $D_{subtrain}$ and treating the remaining 10% of sample as $X_{subtest}, Y_{subtest}$.

### 3.2.3 FINETUNING WITH ADAPTERS

To prevent overfitting and reduce parameter count for each new expert, we only train a small set of new adapter (Houlsby et al., 2019; Bapna et al., 2019; Hu et al., 2021; Liu et al., 2022) parameters, $\psi$, on $p(D|D_{train})$, without modifying in the pretrained transformer's parameters, $\theta$.[5] Specifically, we freeze a pretrained TABPFN transformer, $q_\theta(y|x, D)$. Next, for each downstream dataset, $D_{train}$, we add linear adapter layers (Houlsby et al., 2019), $\mathcal{A}_\psi^{(D_{train})}$, with parameters $\psi$, to form $q_{\theta,\psi}^{(D_{train})}(y|x, D, q_\theta, \mathcal{A}_\psi^{(D_{train})})$. During finetuning, only $\psi$ is optimized. Intuitively, $q_\theta$ encodes the handcrafted prior, $p(D|\phi)p(\phi)$, and $\mathcal{A}_\psi^{(D_{train})}$ encodes the bootstrapped prior, $p(D|D_{train})$. We illustrate CAPFN in Figure 1.

## 4 EXPERIMENT SETUP

We evaluate MIXTUREPFN on the recently proposed TABZILLA benchmark (McElfresh et al., 2023). TABZILLA is the largest tabular benchmark, with 36 hardest datasets out of 176 tabular classification datasets and 19 baseline algorithms, covering both deep learning and GBDTs. The benchmark covers a diverse range of dataset properties, in number of samples, number of features, and feature distributions. We choose this benchmark as it considers a larger range of dataset properties than other existing benchmarks (Grinsztajn et al., 2022; Kadra et al., 2021; Gorishniy et al., 2022). MIXTUREPFN's goal is to (1) improve TABPFN* (McElfresh et al., 2023), which randomly samples $B$ training pairs so that TABPFN (Hollmann et al., 2022) runs on larger datasets, and (2) outperform both GBDTs (Chen & Guestrin, 2016; Prokhorenkova et al., 2018; Ke et al., 2017) which were found state-of-the-art by TABZILLA, and recent deep learning models (Popov et al., 2019; Gorishniy et al., 2021; Arik & Pfister, 2021; Hollmann et al., 2022; Yamada et al., 2020; Yoon et al., 2020; Somepalli et al., 2021; Chen et al., 2022).

---

[5]Adapters are also efficient because only a small number of parameters are updated, $p(\phi)$ is not needed during finetuning, and different downstream datasets share a common pretrained model.

| Method | Mean ± Std Rank↓ | Median Rank↓ | Min Rank↓ | Max Rank↓ |
|---|---|---|---|---|
| MIXTUREPFN | **2.75 ± 1.94** | **2** | **1** | 9 |
| MIXTUREPFN (KNNv2) | 4.00 ±1.56 | 4 | 2 | **7** |
| CatBoost | 4.70 ±3.16 | 6 | **1** | 9 |
| MIXTUREPFN (CaPFN w. Full FT) | 4.85 ±2.03 | 4.5 | 2 | 9 |
| XGBoost | 4.95 ±3.17 | 6 | **1** | 10 |
| MIXTUREPFN (-CaPFN) | 5.10 ±1.12 | 5 | 3 | 8 |
| MIXTUREPFN (-CaPFN-MICP) = TABPFN* | 5.30 ±1.59 | 5 | 2 | 9 |
| MIXTUREPFN (KNNv1) | 7.25 ±3.70 | 9 | **1** | 10 |
| MLP-rtdl | 7.25 ±2.79 | 8 | **1** | 10 |
| MLP | 8.85 ±0.99 | 9 | 7 | 10 |

Table 2: Ablation table results. MIXTUREPFN (KNNv1) and MIXTUREPFN (KNNv2) replace MICP with a scalable variant of KNN-Prompting. MIXTUREPFN (CaPFN w. Full FT) uses full finetuning instead of adapters. MIXTUREPFN (-CaPFN) and MIXTUREPFN (-CaPFN-MICP) remove each component iteratively, where MICP is replaced by random sampling.

### 4.0.1 EVALUATION PROTOCOL

Since TABZILLA restricts the total runtime to 10 hours, not all algorithms run on the same datasets. To ensure a fair comparison[6], we evaluate MIXTUREPFN and baselines using (1) Mean Rank, TABZILLA's original metric, (2) Wilcoxon-Signed Rank Test to check for statistical significance, and (3) Condorcet voting (Gehrlein & Valognes, 2001; Wang et al., 2012), which aggregates pairwise comparisons between each algorithm on each dataset.

We introduce the Condorcet voting to deal with cases where not all algorithms run on all datasets. TABZILLA 's Mean Rank metric is not ideal, because it considers either a subset of datasets or subset of algorithms. This throws away valuable information, since the relative ranking between algorithms on the excluded datasets are ignored. Incomplete ranked-choice voting is well-studied problem in political science (Gehrlein & Valognes, 2001), where voters (datasets) rank candidates (algorithms) relative to each other. Condorcet voting ensures both that every voter's (dataset's) rankings are accounted for and that no candidate (algorithm) or voter (dataset) are artificially excluded, by decomposing full rankings into pair-wise rankings. An algorithm is the Condorcet winner if it wins all pairwise comparisons averaged across all datasets.

## 5 RESULTS

### 5.1 MIXTUREPFN: STATE-OF-THE-ART PERFORMANCE

As shown in Table 1, MIXTUREPFN achieves state-of-the-art performance on TABZILLA across 36 datasets and 19 baseline algorithms **both using TabZilla's original metric, Mean Rank, and as the Condorcet winner, beating all other baselines in pairwise comparisons**. MIXTUREPFN is followed by GBDTs (Chen & Guestrin, 2016; Prokhorenkova et al., 2018), then TABPFN* (McElfresh et al., 2023), then deep learning algorithms (Chen et al., 2022; Gorishniy et al., 2021; Somepalli et al., 2021). These results corroborate recent findings (McElfresh et al., 2023; Grinsztajn et al., 2022) that most deep learning algorithms fail on general tabular datasets. MIXTUREPFN achieves its state-of-the-art results by scaling TABPFN*'s impressive performance to larger datasets. We provide additional metrics in the Appendix. To understand what dataset regimes each algorithm performs best at, we evaluate MIXTUREPFN's rankings w.r.t. dataset properties.

**MIXTUREPFN substantially improves the scalability of TABPFN and TABPFN*.** As show in Figure 2a, unlike TABPFN which encounters memory bottlenecks on datasets with $> 3000$ samples, MIXTUREPFN successfully runs on all said datasets. As show in Figure 2a, MIXTUREPFN substantially improves the performance of TABPFN* by improving how samples are chosen for the "prompt" and training on the downstream dataset.

---

[6]We cannot follow the same experimental settings as the October revision of TABZILLA because they are unfair, as mentioned in a recent Github issue (git, 2024a). Further details are in the Appendix.

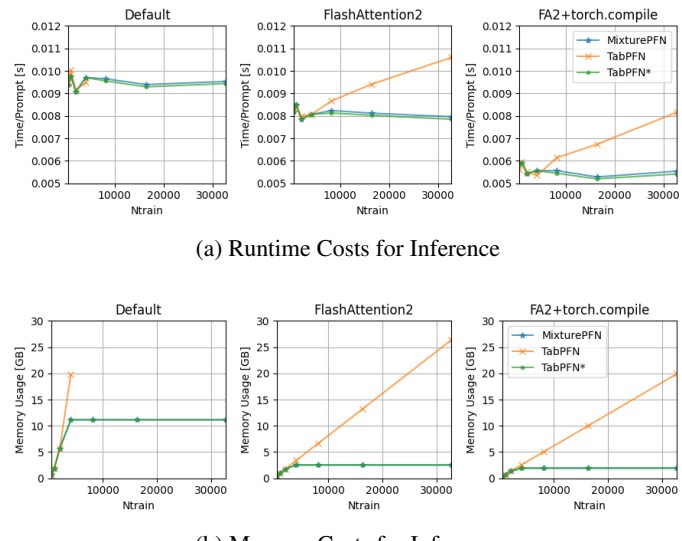

(a) Runtime Costs for Inference

(b) Memory Costs for Inference

Figure 4: Computation Costs for attention compute with different hardware optimizations.

**MIXTUREPFN encounters no performance deterioration w.r.t. dataset size.** As shown in Figure 2b, unlike TABPFN*, whose performance deteriorates w.r.t dataset size, MIXTUREPFN's performance compared to the next best baseline is not correlated with dataset size. Hence, MIXTUREPFN is necessary to scale TABPFN*'s impressive performance on to larger datasets.

**MIXTUREPFN is robust to irregular datasets.** We measure the irregularity of datasets using the standard deviation of the kurtosis of all features. Deep learning algorithms are especially susceptible to irregular datasets with uninformative or heavy-tail features (Grinsztajn et al., 2022). As shown in Figure 2c, MIXTUREPFN is the Top-1 algorithms on datasets with both high and low kurtosis standard deviation. Because it is finetuned on downstream datasets, MIXTUREPFN is robust to dataset irregularity.

**MIXTUREPFN works better with fewer features.** As shown in Figure 2c, MIXTUREPFN loses against baselines on datasets with a large number of features. PFN transformers are known to face scalability challenges with number of features (Hollmann et al., 2022), due to their handling of column order invariance. We believe better tokenization practices and feature selection can improve feature size scalability, and leave such exploration to future work.

**MIXTUREPFN's state-of-the-art performance is statistically significant**. Specifically, we run the Wilcoxon Signed-Rank test with $p < 0.05$ comparing the Top-10 Condorcet algorithms from Table 1 across their 22 shared datasets and 10 cross-validation splits. As shown in Figure 3, MIXTUREPFN's state-of-the-art performance is statistically significant.

## 5.2 ABLATION STUDY

**Both MICP and CAPFN contribute to MIXTUREPFN state-of-the-art results.** We perform ablation studies for MICP and CAPFN against common GBDTs and deep learning models across 10 shared datasets. As shown in Table 2, each component of the model, MICP and CAPFN, contributes to achieving state-of-the-art results. MICP helps by efficiently choosing an effective context for the "prompt". CAPFN helps by aligning the dataset prior through finetuning the PFN on MICP's prompting policy. Because overfitting is a well-known issue for deep learning models tackling tabular data (Kadra et al., 2021; Grinsztajn et al., 2022), adapters are a key component to ensure CAPFN aligns the pretrained TABPFN transformer with the downstream data, without destroying its pretraining prior, $p(D|\phi)p(\phi)$.

**Under the same GPU resources, KNN-Prompting is much less effective than MICP on tabular datasets**. As described in Section 2.1.3, KNN-Prompting does not support TABPFN-style batching. To empirically verify that MICP improves KNN-Prompting, we replace MICP in MIXTUREPFN

with 2 KNN-Prompting variants: **MIXTUREPFN (KNNv1)**: Because each prompt contains at most $B + N_{batch}$ tokens, we batch KNN-Prompts by considering $B/N_{batch}$ nearest neighbors instead of $B$-nearest neighbors; **MIXTUREPFN (KNNv2)**: Because LLM-batching fails due to TABPFN's ensembling overhead, we remove the ensembling procedure and run KNN-Prompting following LLM-batching (Liu et al., 2021), as described in Section 2.1.3. As shown in Table 2, both KNN-Prompting variants perform substantially worse than MIXTUREPFN because they compromise an essential component of PFN, either prompt size or ensembling, for performing efficient inference. The relative rankings suggest prompt size matters more than ensembling.

## 5.3 RUNTIME

Given the efficiency effectiveness tradeoffs listed in Section 2.1.3, we provide real runtime analysis between TABPFN, TABPFN*, and MIXTUREPFN. Specifically, we set batch size to 32, ensemble size to 32, feature count to 100, and time the transformer forward pass combined along with any preprocessing costs, namely MIXTUREPFN routing. We

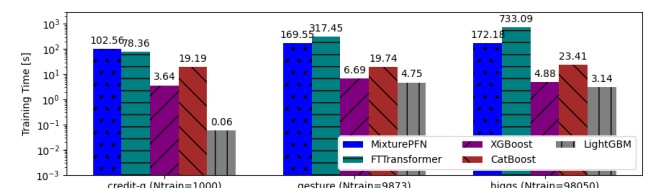

Figure 5: Train time costs for MIXTUREPFN.

report median runtime and memory consumption across 8 runs to remove outliers. Note, recent hardware-level optimizations, specifically FlashAttention (Dao, 2023) and torch compile, greatly improve real runtimes of transformer models. For fair comparison, we improve TABPFN with FlashAttention (Dao, 2023) and torch compile. **MIXTUREPFN is more runtime and memory efficient than full-context TABPFN independent of hardware optimizations** by only attending to training samples in the local neighborhood of grouped test samples. As seen in Figure 4, MIXTUREPFN incurs the same memory costs as TABPFN* and nearly the same runtime costs as TABPFN* due to highly optimized nearest-neighbor search algorithms (Douze et al., 2024).

We quantify the efficiency effectiveness tradeoff of CAPFN finetuning by computing the runtime of training. Note, Section 5.2 shows that finetuning costs do substantially boost performance. We compare MIXTUREPFN 's training times to XGBOOST, CatBoost, Saint, LightGBM, and FTTransformer. We report training runtime for one hyperparameter setup of one cross-validation fold over datasets of 3 varying sizes, and report the median runtime/memory consumption across 3 runs to remove outliers. As shown in Figure 5, **MIXTUREPFN takes roughly the same amount of training time per expert as other deep learning algorithms, like FTTransformer.** While decision tree algorithms still run faster than deep learning algorithms, MIXTUREPFN substantially improves the trade-offs between tree-based and deep learning algorithms through its impressive performance.

## 6 LIMITATIONS

As shown in Section 5.1, MIXTUREPFN successfully improves TABPFN's scalability w.r.t. dataset size to achieve state-of-the-art results on the TABZILLA benchmark. However, we notice that TABPFN* does not scale well with feature and label count, relying on ensembling to capture feature and label order invariance. Scaling TABPFN* to datasets with large number of features and labels can further push ICL performance for tabular learning. While this work covers a large number of diverse datasets, we do not cover huge datasets with billions of samples (Zhu et al., 2023; Yang et al., 2023). We leave such investigation to future work.

## 7 CONCLUSION

In this work, we provide a scalable framework for In-Context Learning (ICL) on tabular datasets. To efficiently construct effective ICL "prompts", we propose routing test samples through a Sparse Mixture of In-Context Prompters, MICP. To align the PFN with the inference-time datasets, we propose a novel finetuning policy using bootstrapping, CAPFN. Our framework, MIXTUREPFN, achieves state-of-the-art performance against 19 deep learning and tree-based baselines across 36 general tabular datasets.

# 8 ETHICS STATEMENT

This paper presents work whose goal is to advance the field of tabular and in-context learning. Our work reaches state-of-the-art tabular classification accuracy, which has broad positive impact for many industries using relational databases and tabular datasets. We hope our impressive results inspire further research into PFNs and ICL for tabular learning. Our work is built on large transformer models, which are known to hallucinate in the natural language domain. While we observe no such behavior on our tabular datasets, we will open source our code, such that practitioners can plug in their own safe transformer models. We feel there are not any other noteworthy negative societal impacts.

# 9 ACKNOWLEDGEMENTS

This work was done at an internship at Uber and was partially supported by NSF 2211557, 1937599, 2119643, 2303037, 2312501, 2200274, 2106859, NIH U54HG012517, U24DK097771, NASA, SRC JUMP 2.0 Center, Amazon Research Awards, Snapchat Gifts, and Optum AI.

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

## A  RELATED WORK

### A.1  TABULAR LEARNING ALGORITHMS

Early tabular learning algorithms are based off decision trees, utilizing boosting, feature encoding, and ensembling (Shwartz-Ziv & Armon, 2022; Borisov et al., 2022; Chen & Guestrin, 2016). Early deep learning algorithms are inspired by decision trees, making them end-to-end learnable (Popov et al., 2019; Katzir et al., 2020; Hazimeh et al., 2020; Somepalli et al., 2021; Arik & Pfister, 2021); however, more thorough benchmarks find decison trees produce more reliable results. Hyperparameter tuning (Kadra et al., 2021) and inductive bias (Grinsztajn et al., 2022) were identified as key weaknesses in deep learning algorithms. Recent works focus on optimizing transformer models for specialized datasets (Huang et al., 2020; Gorishniy et al., 2021; 2022), and improving decision tree optimization (Joseph & Raj, 2022), or Bayesian learning (Hollmann et al., 2022; Schäfl et al., 2022; Feuer et al., 2023) for small datasets. With the rise of LLMs, pretrained tabular learning models also achieve impressive performance at the cost of billions of training points (Yang et al., 2023; Zhu et al., 2023). Of these methods, Prior-Fitted Networks (Feuer et al., 2023) were identified as a promising direction in recent benchmarks among deep learning approaches for general tabular learning problems (McElfresh et al., 2023). Most recently, TuneTables (Feuer et al., 2024) appeared on arxiv using sketching to also scale TABPFN at inference; however, purely inference-time techniques will always have some performance gap due to distribution shift, which our work addresses.

### A.2  GRADIENT-BOOSTED DECISION TREES

Gradient-boosted decision trees (GBDTs) remain the preferred algorithm of tabular learning practitioners (Chen & Guestrin, 2016; Prokhorenkova et al., 2018). Deep learning algorithms (Popov et al., 2019; Gorishniy et al., 2021; Somepalli et al., 2021; Arik & Pfister, 2021; Yamada et al.,

2020; Yoon et al., 2020) fail on larger benchmarks considering different numbers of samples, numbers of features, feature types, feature distributions, and numbers of labels (McElfresh et al., 2023; Shwartz-Ziv & Armon, 2022). Because GBDTs utilize boosted gradients and are not rotationally invariant before training, GBDT learning algorithms have a better inductive bias than Stochastic Gradient Descent-based (SGD-based) deep learning algorithms Grinsztajn et al. (2022). Thus, GBDTs achieve state-of-the-art performance on medium to large datasets with 3,000 to 1,000,000 samples and competitive performance on smaller datasets McElfresh et al. (2023); Grinsztajn et al. (2022).

## A.3 PRIOR FITTED NETWORKS

Prior Fitted Networks (PFN) (Müller et al., 2021) approximates Bayesian inference using a data prior, where a parameterized model is trained to minimize the KL-Divergence between it and the posterior predictive distribution. The proof for PFNs is derived from meta-learning (Gordon et al., 2018). PFNs fall under In-Context Learning (Brown et al., 2020), as the entire training dataset is fed to the model during inference. Hence, PFNs effectively learn the learning algorithm. This approach is particularly effective on tabular data (McElfresh et al., 2023), where the data prior effectively regularizes model predictions. TABPFN (Hollmann et al., 2022) is a PFN model specific to tabular data that achieves state-of-the-art performance on small datasets. Unlike preliminary works on scaling TABPFN (Feuer et al., 2023), that benchmark KMeans and Coreset "promting", we propose a sparse mixture of KNN prompters, capable of forming scalable batched "prompts" and a novel finetuning protocol for the mixture of prompters. Furthermore, our approach not only improves TABPFN results on large datasets, but achieves state-of-the-art performance across tabular benchmarks (McElfresh et al., 2023).

## A.4 MIXTURE OF EXPERTS

Sparsely gated MoE (Shazeer et al., 2017; Lewis et al., 2021) shows a significant improvement in model capacity, training time, and accuracy with a gating mechanism. An expert is a sub-network, which better learns to predict similar data points. A gating mechanism, learnable or non-learnable method, decides to route each data point to the most suited experts (Shazeer et al., 2017). Switch transformers (Fedus et al., 2022) is a learning to route approach, where it assigns one data point to only one expert, instead of top-k, which reduces computation, while preserving accuracy. However, learnable routing methods require auxiliary load balancing loss function, and further tuning. In (Roller et al., 2021), a non-learnable routing method is proposed, which uses a hashing method to assign similar data points to similar experts. They show that this procedure can be better or be competitive with learnable routing MoE methods. However, a hashing method is not necessarily flexible in assigning data points to suitable experts, as it can cause data skewness and choice of hashing function is sensitive to the downstream task. Inspired by these works, we proposed a non-learnable routing mechanism which assigns one data point to one expert, and our routing method finds the most suitable expert based on similarity of data points with a K-Means clustering method. Also, we used experts with shared weights as opposed to general MoE in most of the previous works, as the prompts, not the model, contains training examples for In-Context Learning. We highlight tackling MoE in the context of prompting is a newly emerging research interest (Anonymous, 2024), of which we are among the first.

## A.5 EFFICIENT TRANSFORMERS

Long sequence inputs have long been studied by the efficient transformer community. Several linear time and space complexity transformers have been proposed (Katharopoulos et al., 2020; Wang et al., 2020; Choromanski et al., 2020; Qin et al., 2022), primarily be SVD decomposing the attention computation. While efficient transformers can help scale the base PFN model, linear complexity is too expensive for the scale of tabular data in industry. Furthermore, many approaches require finetuning on downstream data (Katharopoulos et al., 2020; Wang et al., 2020) which is nontrivial for PFN models. Constant time transformers (Zaheer et al., 2020; Bulatov et al., 2023; Chowdhery et al., 2023) exploit the sequential nature of text data. These methods also do not apply to PFNs for tabular data, as tabular training data is not inherently sequential. Hence, technologies outside of efficient transformers are needed to effectively scale PFNs for tabular learning.

## A.6 IN-CONTEXT LEARNING

In-Context Learning (ICL) prompts transformers with training examples prepended to the desired query (Brown et al., 2020). Several works attempt to prompt engineer scalable in-context examples for better downstream performance (Hao et al., 2022), among which K-Nearest Neighbors emerge as a reliable choice (Liu et al., 2021; Xu et al., 2023). However, ICL for LLMs consider queries one-at-a-time instead of in batch fashion, prompts are encoded as natural language, and in-context examples can come from a large corpus of natural lanaguage data (Brown et al., 2020). These properties are not afforded to PFN-style ICL, where inference is directly run on training set tokens. In addition to scaling "prompts", LLMs can also be finetuned on ICL examples to reach better performance (Thoppilan et al., 2022; Wei et al., 2021). However, this is due to LLM pretraining objectives misaligning with the ICL task. Such misalignment is not as obvious in the PFN case, where the transformer is directly trained on the ICL task (Müller et al., 2021). Hence, we propose Sparse Mixture of In-Context Prompters to support batching and our novel bootstrapping algorithm to finetune PFN models. In the spirit of multi-modal models (Radford et al., 2021; Xu et al., 2022), TABPFN (Hollmann et al., 2022) extends ICL techniques from natural language LLMs to tabular data.

## B    MATH NOTATIONS

We summarize our math notations below:

- $B$: Number of training samples in prompt
- $K$: Number of experts
- $D$: A generic dataset
- $D_{train}$: The training dataset
- $X_{test}$: The test samples
- $X_{batch}$: A batch of testing data
- $C$: Number of classes
- $q_\theta$: PFN Model
- $A_\psi^{D_{train}}$: PFN Model Adapters trained on $D_{train}$
- $\theta$: PFN Model Parameters
- $\psi$: PFN Model Adapter Parameters
- $\phi$: PFN hypothesis mechanism
- $\mathcal{R}$: Router mapping input test points to 1 of $K$ Prompters
- $\mathcal{T}_k$: The $k$-th Prompter
- $D_{cluster}^{(k)}$: The $k$-th K-Means cluster of training samples
- $D_{prompt}^{(k)}$: The $k$-th Prompter's training samples context
- $(X_{batch}, D_{prompt}^{(k)})$: The $k$-th Prompter's "prompt"
- $NNS(\cdot|\cdot)$: Nearest Neighbor Search Algorithm
- $KNN(\cdot|\cdot, \cdot)$: K-Nearest Neighbors Algorithm
- $KMeans(\cdot)$: K-Means Algorithm
- $Sample(\cdot)$: Random Sampling
- $N_{train}$: Full training dataset size
- $N_{test}$: Full testing dataset size
- $N_{batch}$: Batch size
- $\gamma$: Single hyperparameter trading off performance and efficiency.

## C   NONZERO OVERLAP PROOF

We prove Theorem 3.1.1 here.

First we prove $|D_{cluster}^{(k)}| \leq B \implies D_{cluster}^{(k)} \subseteq \text{KNN}(x_{center}^{(k)}|D_{train}, B)$:

By KMeans definition,

$$d(x_{center}^{(k)}, x_{train}^{(i)}) < d(x_{center}^{(k)}, x_{train}^{(j)}) \implies (x_{train}^{(j)} \in D_{cluster}^{(k)} \implies x_{train}^{(i)} \in D_{cluster}^{(k)}),$$

$$\implies \exists \tau_{KMeans}^{(k)} : d(x_{center}^{(k)}, x_{train}^{(i)}) < \tau_{KMeans}^{(k)} \implies x_{train}^{(i)} \in D_{cluster}^{(k)}$$

By KNN definition,

$$\text{KNN}(x|D_{train}, B) = \{x_{train}^{(i)} : d(x_{center}^{(k)}, x_{train}^{(i)}) < \tau_{KNN}(x|D_{train})\},$$

where $|\{x_{train}^{(i)} : d(x_{center}^{(k)}, x_{train}^{(i)}) < \tau_{KNN}(x|D_{train})\}| = B$

Given $|D_{cluster}^{(k)}| \leq B$,

$$\implies |D_{cluster}^{(k)}| \leq |\text{KNN}(x_{center}^{(k)}|D_{train}, B)|$$

$$\implies \text{KNN}(x_{center}^{(k)}|D_{train}, B) = D_{cluster}^{(k)} \cup \{x_{train}^{(i)} : \tau_{KMeans}^{(k)} \leq d(x_{center}^{(k)}, x_{train}^{(i)}) < \tau_{KNN}(x|D_{train})\}$$

$$\implies D_{cluster}^{(k)} \subseteq \text{KNN}(x_{center}^{(k)}|D_{train}, B)$$

Next, we prove Theorem 3.1.1:

Given $|D_{cluster}^{(k)}| \leq B \ \forall k \in [0, ..., K-1]$ and $\mathcal{R}^*(x_{train}^{(i)}) = p : x_{train}^{(i)} \in D_{cluster}^{(k)}$,

$$\implies G(k) = 1$$

$$\implies x_{train}^{(i)} \in \text{KNN}(x_{train}^{(i)}|D_{train}, B) = \text{KNN}(x_{test}^{(i)}|D_{train}, B)$$

$$\implies x_{train}^{(i)} \in D_{cluster}^{(\mathcal{R}^*(x_{train}^{(i)}))} \subseteq \text{KNN}(x_{center}^{(\mathcal{R}^*(x_{train}^{(i)}))}|D_{train}, B) = D_{prompt}(\mathcal{R}^*(x_{train}^{(i)}))$$

$$\implies x_{train}^{(i)} \in D_{prompt}(\mathcal{R}^*(x_{train}^{(i)})) \cap \text{KNN}(x_{test}^{(i)}|D_{train}, B)$$

$$\implies |D_{prompt}(\mathcal{R}^*(x_{train}^{(i)})) \cap \text{KNN}(x_{test}^{(i)}|D_{train}, B)| \geq |\{x_{train}^{(i)}\}| = 1 \ \forall i \in [0, ..., N_{train}-1]$$
$\forall D_{train}$.

## D   SIMILARITY AND DIFFERENCES COMPARED TO LARGE LANGUAGE MODELS

We illustrate the difference between ICL on PFNs and LLMs in Figure 6.

## E   SUPPORT SET AND KNN PROMPTING.

In Figure 15, we point out one key difference between KNN-Prompting with TABPFN (Hollmann et al., 2022) and with ICL on LLMs (Liu et al., 2021): LLMs support batching multiple test pairs across multiple "prompts" where each "prompt" contains the nearest neighbors to 1 test point. In contrast, TABPFN requires multiple test cases in 1 "prompt", which necessitates techniques like MICP to route a batch of test points to the same "prompt". As shown in Section 5.2, TABPFN's more efficient prompts achieve better performance under same GPU constraints as ICL for LLM batched prompts. We leave application of MICP to ICL and LLMs as future work.

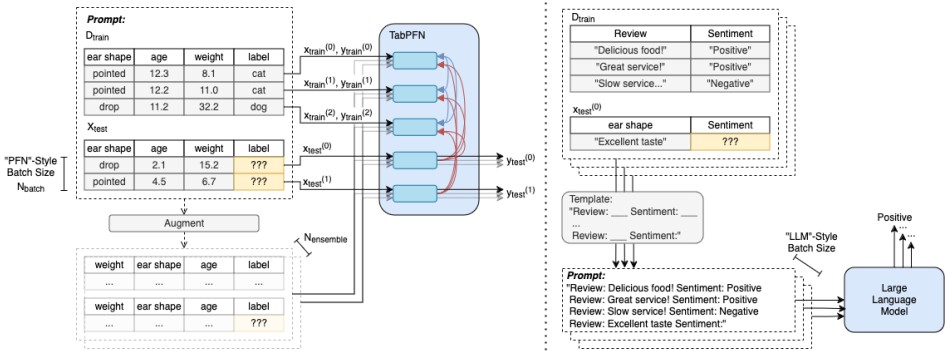

Figure 6: We highlight the differences between In-Context Learning (ICL) on Prior Fitted Networks (ex. TABPFN), left, and Large Language Models (LLMs), right. TABPFN treats training data as tokens (where each token is a concatenation of feature and label), whereas LLMs use templates to convert training data into natural language prompts. TABPFN uses an attention pattern (blue and red arrows) supporting batch inference, whereas LLMs use generic encoder-decoder or decoder-only setups. TABPFN are pretrained on Equation 1, whereas LLMs are pretrained separately.

## F TIME AND SPACE COMPLEXITY DETAILS

**Router and prompter initialization takes** $\mathcal{O}(tN_{train}K + (N_{train} + KB)logN_{train})$ **time and** $\mathcal{O}(N_{train} + KB)$ **space complexity and is done once before inference.** For initialization, K-Means with $t$-iterations takes $\mathcal{O}(tN_{train}K)$ time and $\mathcal{O}(K)$ space. To perform efficient nearest neighbor queries, we use the ball-tree algorithm over the training dataset and cluster centers, which takes $\mathcal{O}(N_{train}log(N_{train}))$ time and $\mathcal{O}(N_{train})$ space. Using ball-tree KNN queries, constructing each ICP support set takes $\mathcal{O}(Blog(N_{train}))$ time and $\mathcal{O}(B)$ space.

**Routing takes** $\mathcal{O}(log(K))$ **time and** $\mathcal{O}(1)$ **space complexity, using efficient nearest neighbor search with ball-tree for each test sample.** Router overhead is practically overcome via highly-optimized NNS implementations (Douze et al., 2024), which scale $K$ to the billions.

**PFN transformer inference takes** $\mathcal{O}(B^2 + BN_{batch})$ **time and space complexity, as MICP prompts contain at most** $B$ **training samples and** $N_{batch} = |X_{batch}|$ **testing samples.** Note, unlike purely KNN-based prompting, MICP supports batched computation to further amortize PFN inference cost.

## G CHOOSING BASELINE DATASETS AND ALGORITHMS

We chose our datasets from the TABZILLA benchmark, which curates 36 of the hardest 176 considered datasets across 19 algorithms. As noted in Section 4.0.1, not all algorithms run on all datasets. We note which datasets are shared by all algorithms, Top-10 algorithms, and Top-5 algorithms in Table 3. We provide the number of datasets each algorithm successfully runs on in Table 11. We provide dataset names and statistics in Table 19.

The 2 datasets MIXTUREPFN and TABPFN* fails on contain ¿10 classes, which is not currently supported by the pretrained TABPFN. However, as seen in Figure 16 and described in Section G.1, we highlight that the 34 datasets MIXTUREPFN and TABPFN* successfully run on **cover the full range of dataset properties: number of features, number of samples, and dataset irregularity.** Specifically, our 34 datasets include the ones with the least and most number of samples, the least and most number of features, and the least and most irregularities. Because the focus of this work is to scale TABPFN to datasets with more number of samples, we leave extending TABPFN to more number of classes as future work.

### G.1 BASELINE DATASETS

We provide dataset statistics in Table 19. As shown, our considered datasets cover a wide range of dataset properties in number of features, number of samples, and std. kurtosis. As shown, MIX-

| Subset | Considered Algorithms | Considered Datasets |
|--------|----------------------|---------------------|
| ALL | MixturePFN, CatBoost, TabPFN*, XGBoost, ResNet, FTTransformer, LightGBM, SAINT, NODE, MLP-rtdl, RandomForest, TabNet, MLP, DecisionTree, LinearModel, STG, VIME, KNN, DANet, SVM | ada-agnostic, australian, balance-scale, colic, credit-approval, elevators, heart-h, jasmine, kc1, lymph, mfeat-fourier, mfeat-zernike, monks-problems-2, phoneme, profb, qsar-biodeg, socmob, speeddating, splice, vehicle |
| Top-10 | MixturePFN, CatBoost, TabPFN*, XGBoost, ResNet, LightGBM, SAINT, DANet, FTTransformer, SVM, | ada-agnostic, artificial-characters, australian, balance-scale, colic, credit-approval, elevators, gesturephasesegmentationprocessed, heart-h, jasmine, kc1, lymph, mfeat-fourier, mfeat-zernike, monks-problems-2, phoneme, profb, qsar-biodeg, socmob, speeddating, splice, vehicle |
| Top-5 | MixturePFN, CatBoost, TabPFN*, XGBoost, SAINT | ada-agnostic, artificial-characters, australian, balance-scale, colic, credit-approval, electricity, elevators, albert, gesturephasesegmentationprocessed, heart-h, higgs, jasmine, jungle-chess-2pcs-raw-endgame-complete, kc1, lymph, mfeat-fourier, mfeat-zernike, monks-problems-2, phoneme, profb, qsar-biodeg, socmob, speeddating, splice, vehicle |

Table 3: Datasets and algorithms considered in each Top-K subset. For fair evaluation (git, 2024a), we only consider the shared set of datasets all algorithms run on in the Top-K subsets. By considering different subsets, we evaluate MIXTUREPFN against more or less algorithms and datasets. 20 datasets are shared by all 20 algorithms. 22 datasets are shared by Top-10 algorithms. 25 datasets are shared by Top-5 algorithms. MIXTUREPFN achieves the best mean rank among all Top-K subsets.

| Method | Condorcet Statistics | | | |
|--------|---------|--------|-------|----------|
| | #Votes↑ | #Wins↑ | #Ties | #Losses↓ |
| MixturePFN | **464** | **19** | 0 | **0** |
| CatBoost | 462 | 18 | 0 | 1 |
| XGBoost | 448 | 16 | 1 | 2 |
| SAINT | 402 | 15 | 0 | 4 |
| ResNet | 367 | 14 | 2 | 3 |
| LightGBM | 360 | 12 | 0 | 7 |
| FTTransformer | 345 | 13 | 1 | 5 |
| TabPFN* | 330 | 11 | 0 | 8 |
| NODE | 304 | 11 | 0 | 8 |
| DANet | 300 | 10 | 1 | 8 |
| RandomForest | 299 | 12 | 1 | 6 |
| MLP-rtdl | 256 | 8 | 0 | 11 |
| SVM | 236 | 6 | 0 | 13 |
| TabNet | 231 | 4 | 0 | 15 |
| MLP | 229 | 7 | 0 | 12 |
| STG | 188 | 5 | 0 | 14 |
| DecisionTree | 155 | 2 | 0 | 17 |
| LinearModel | 144 | 3 | 0 | 16 |
| KNN | 112 | 0 | 0 | 19 |
| VIME | 93 | 1 | 0 | 18 |

Table 4: MIXTUREPFN is the Condorcet winner across 36 datasets against 19 baseline algorithms. We rank algorithms based on their accuracies.

| Method | All Algorithms (Log Likelihood) | | | |
|--------|----------------------|----------------|-------------|-------------|
| | Mean ± Std Rank↓ | Median Rank↓ | Min Rank↓ | Max Rank↓ |
| MixturePFN | **2.350 ± 1.824** | **1.0** | **1.0** | **6.0** |
| TabPFN* | 4.550 ± 2.747 | 4.5 | 2.0 | 13.0 |
| CatBoost | 4.900 ± 4.158 | 3.0 | **1.0** | 14.0 |
| XGBoost | 5.500 ± 4.621 | 4.0 | **1.0** | 16.0 |
| SAINT | 8.300 ± 5.367 | 7.0 | **1.0** | 19.0 |
| ResNet | 8.400 ± 3.262 | 8.5 | 3.0 | 15.0 |
| FTTransformer | 8.600 ± 3.541 | 8.5 | 3.0 | 15.0 |
| DANet | 9.050 ± 3.369 | 8.5 | 3.0 | 17.0 |
| LightGBM | 9.150 ± 4.351 | 10.5 | 2.0 | 16.0 |
| MLP-rtdl | 10.800 ± 5.046 | 10.5 | 2.0 | 20.0 |
| SVM | 11.300 ± 4.766 | 11.0 | 2.0 | 19.0 |
| RandomForest | 11.600 ± 4.443 | 12.5 | 4.0 | 18.0 |
| STG | 11.900 ± 4.549 | 12.5 | 4.0 | 19.0 |
| LinearModel | 12.400 ± 4.652 | 12.5 | 5.0 | 20.0 |
| NODE | 13.350 ± 3.410 | 13.5 | 7.0 | 18.0 |
| TabNet | 13.550 ± 5.296 | 14.0 | 2.0 | 20.0 |
| MLP | 13.700 ± 3.621 | 14.0 | 4.0 | 19.0 |
| VIME | 15.350 ± 3.851 | 17.0 | 6.0 | 19.0 |
| DecisionTree | 16.800 ± 3.881 | 18.5 | 7.0 | 20.0 |
| KNN | 18.450 ± 1.936 | 19.0 | 14.0 | 20.0 |

Table 5: MIXTUREPFN achieves the top mean rank w.r.t. Log Likelihood across 20 datasets where all algorithms successfully run.

| Method | All Algorithms (Accuracy) | | | |
|---|---|---|---|---|
| | Mean ± Std Rank↓ | Median Rank↓ | Min Rank↓ | Max Rank↓ |
| MixturePFN | **3.950 ± 3.570** | **3.0** | **1.0** | **13.0** |
| TabPFN* | 5.500 ± 4.056 | 5.5 | **1.0** | 15.0 |
| CatBoost | 6.350 ± 5.313 | 4.0 | **1.0** | 17.0 |
| XGBoost | 7.100 ± 5.234 | 5.5 | **1.0** | 18.0 |
| ResNet | 7.600 ± 4.017 | 6.5 | **1.0** | 16.0 |
| FTTransformer | 7.900 ± 4.158 | 7.5 | **1.0** | 17.0 |
| SAINT | 7.950 ± 5.723 | 5.5 | **1.0** | 20.0 |
| LightGBM | 8.750 ± 4.918 | 8.5 | **1.0** | 17.0 |
| NODE | 8.850 ± 3.395 | 9.0 | 3.0 | 15.0 |
| MLP-rtdl | 9.150 ± 5.033 | 9.0 | **1.0** | 18.0 |
| RandomForest | 9.350 ± 4.757 | 8.5 | 4.0 | 19.0 |
| DANet | 10.000 ± 4.405 | 11.0 | 3.0 | 19.0 |
| SVM | 12.250 ± 5.476 | 14.5 | **1.0** | 19.0 |
| TabNet | 13.050 ± 4.780 | 13.5 | 2.0 | 20.0 |
| MLP | 13.100 ± 4.253 | 14.5 | 5.0 | 18.0 |
| LinearModel | 14.100 ± 3.846 | 14.0 | 8.0 | 20.0 |
| DecisionTree | 14.250 ± 4.426 | 15.0 | 3.0 | 20.0 |
| STG | 14.800 ± 4.423 | 16.0 | 4.0 | 20.0 |
| VIME | 16.950 ± 2.854 | 18.0 | 10.0 | 20.0 |
| KNN | 17.400 ± 3.470 | 19.0 | 7.0 | 20.0 |

Table 6: MIXTUREPFN achieves the top mean rank w.r.t. Accuracy across 20 datasets where all algorithms successfully run.

| Method | Top-10 Algorithms (Log-Likelihood) | | | |
|---|---|---|---|---|
| | Mean ± Std Rank↓ | Median Rank↓ | Min Rank↓ | Max Rank↓ |
| MixturePFN | **2.273 ± 1.710** | **1.0** | **1.0** | **6.0** |
| CatBoost | 3.955 ± 2.688 | 3.0 | **1.0** | 10.0 |
| XGBoost | 4.000 ± 2.663 | 3.0 | **1.0** | 10.0 |
| TabPFN* | 4.045 ± 1.965 | 4.0 | 2.0 | 9.0 |
| SAINT | 6.136 ± 2.473 | 6.5 | **1.0** | 10.0 |
| LightGBM | 6.409 ± 2.839 | 7.5 | 2.0 | 10.0 |
| FTTransformer | 6.773 ± 2.235 | 7.0 | 3.0 | 10.0 |
| ResNet | 6.864 ± 1.961 | 7.0 | 3.0 | 9.0 |
| DANet | 7.045 ± 1.988 | 7.0 | 3.0 | 10.0 |
| SVM | 7.500 ± 2.482 | 8.0 | 2.0 | 10.0 |

Table 7: MIXTUREPFN achieves the top mean rank w.r.t. Log Likelihood across 22 datasets where all Top-10 algorithms successfully run.

| Method | Top-10 Algorithms (Accuracy) | | | |
|---|---|---|---|---|
| | Mean ± Std Rank↓ | Median Rank↓ | Min Rank↓ | Max Rank↓ |
| MixturePFN | **3.000 ± 2.256** | **2.0** | **1.0** | **9.0** |
| TabPFN* | 4.318 ± 2.703 | 4.5 | **1.0** | **9.0** |
| CatBoost | 4.591 ± 2.964 | 4.0 | **1.0** | 10.0 |
| XGBoost | 4.773 ± 2.859 | 4.0 | **1.0** | 10.0 |
| ResNet | 5.591 ± 2.103 | 5.5 | **1.0** | **9.0** |
| LightGBM | 5.727 ± 2.847 | 6.5 | **1.0** | 10.0 |
| SAINT | 5.727 ± 2.847 | 5.0 | **1.0** | 10.0 |
| FTTransformer | 5.864 ± 2.282 | 6.0 | **1.0** | **9.0** |
| DANet | 6.955 ± 2.184 | 7.5 | 3.0 | 10.0 |
| SVM | 7.773 ± 2.907 | 9.0 | **1.0** | 10.0 |

Table 8: MIXTUREPFN achieves the top mean rank w.r.t. Accuracy across 22 datasets where all Top-10 algorithms successfully run.

| Method | Top-5 Algorithms (Log Likelihood) | | | |
|---|---|---|---|---|
| | Mean ± Std Rank↓ | Median Rank↓ | Min Rank↓ | Max Rank↓ |
| MixturePFN | **2.000 ± 1.166** | **1.0** | **1.0** | **4.0** |
| XGBoost | 2.760 ± 1.394 | 3.0 | **1.0** | 5.0 |
| CatBoost | 2.880 ± 1.243 | 3.0 | **1.0** | 5.0 |
| TabPFN* | 3.320 ± 1.085 | 3.0 | 2.0 | 5.0 |
| SAINT | 4.040 ± 1.311 | 5.0 | **1.0** | 5.0 |

Table 9: MIXTUREPFN achieves the top mean rank w.r.t. Log Likelihood across 25 datasets where all Top-5 algorithms successfully run.

| Method | Top-5 Algorithms (Accuracy) | | | |
|---|---|---|---|---|
| | Mean ± Std Rank↓ | Median Rank↓ | Min Rank↓ | Max Rank↓ |
| MixturePFN | **2.360 ± 1.292** | **2.0** | **1.0** | **5.0** |
| XGBoost | 2.880 ± 1.395 | 3.0 | **1.0** | **5.0** |
| CatBoost | 2.920 ± 1.354 | 3.0 | **1.0** | **5.0** |
| TabPFN* | 3.000 ± 1.386 | 3.0 | **1.0** | **5.0** |
| SAINT | 3.680 ± 1.406 | 4.0 | **1.0** | **5.0** |

Table 10: MIXTUREPFN achieves the top mean rank w.r.t. Accuracy across 25 datasets where all Top-5 algorithms successfully run.

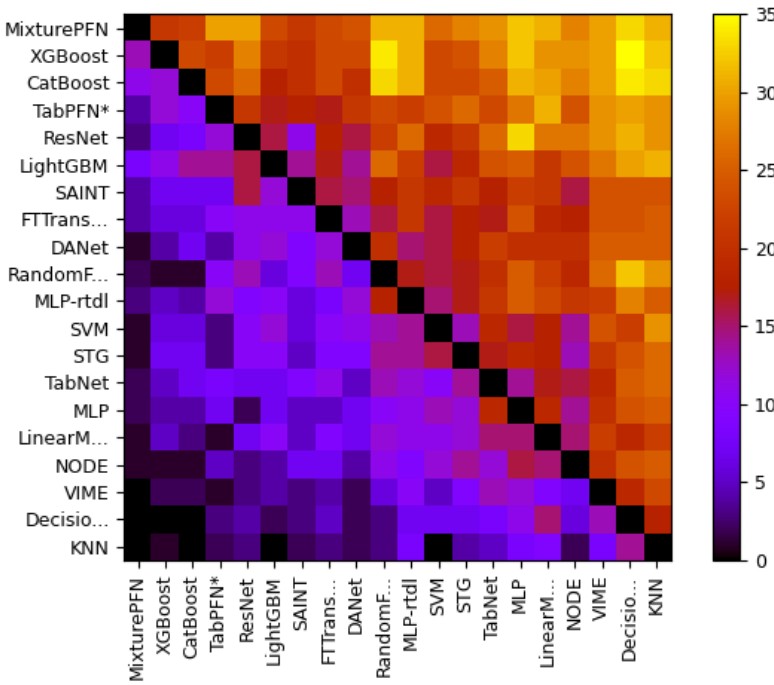

Figure 7: Pairwise comparison matrix for Condorcet voting over the log likelihood metric. Note, MIXTUREPFN is the Condorcet winner.

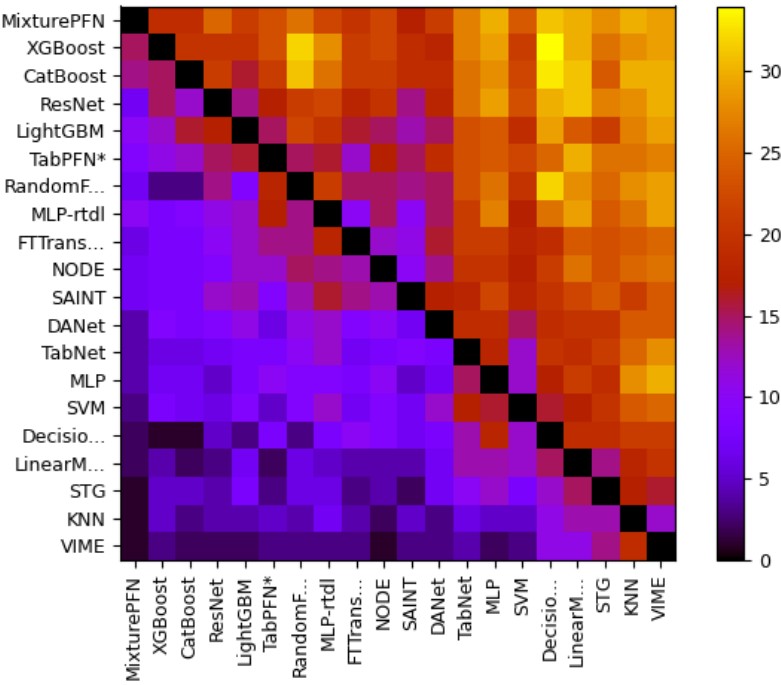

Figure 8: Pairwise comparison matrix for Condorcet voting over the accuracy metric. Note, MIX-TUREPFN is the Condorcet winner. Please refer to Section H for more discussion.

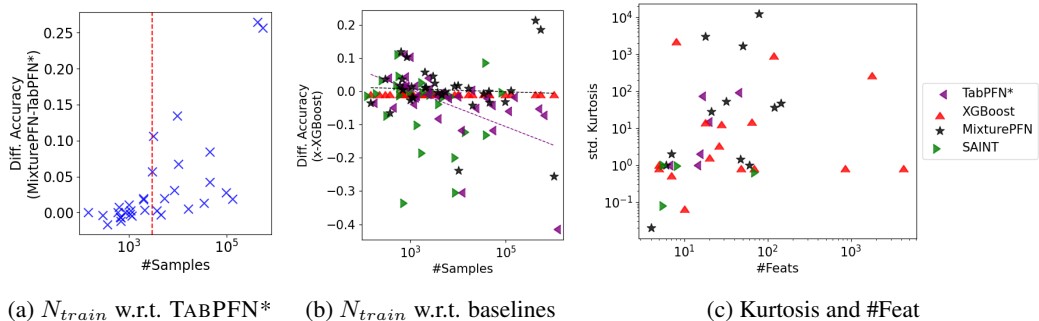

(a) $N_{train}$ w.r.t. TABPFN*    (b) $N_{train}$ w.r.t. baselines    (c) Kurtosis and #Feat

Figure 9: We perform the same sensitivity analysis as Figure 2 in the main text on the accuracy metric.

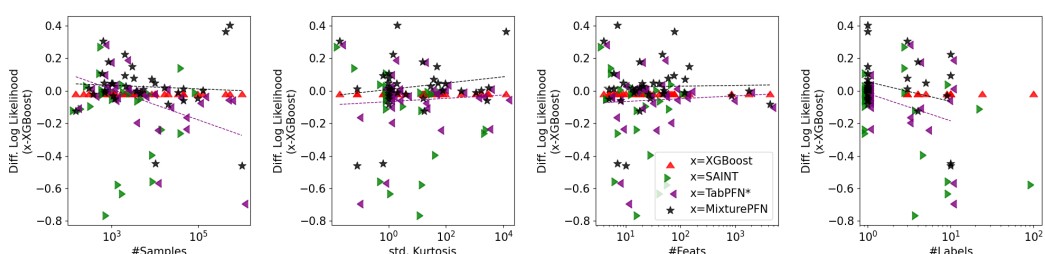

(a) $N_{train}$ w.r.t. baselines    (b) Kurtosis w.r.t. baselines    (c) #Feat w.r.t. baselines    (d) #Class w.r.t. baselines

Figure 10: We perform the same sensitivity analysis as Figure 2 in the main text but across all dataset properties.

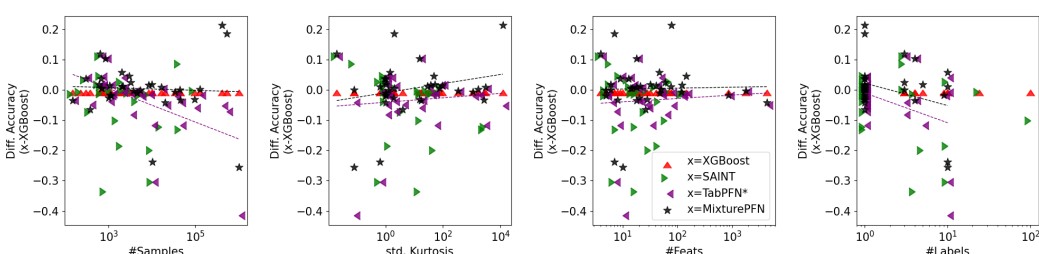

(a) $N_{train}$ w.r.t. baselines    (b) Kurtosis w.r.t. baselines    (c) #Feat w.r.t. baselines    (d) #Class w.r.t. baselines

Figure 11: We perform the same sensitivity analysis as Figure 2 in the main text but across all dataset properties and on the accuracy metric.

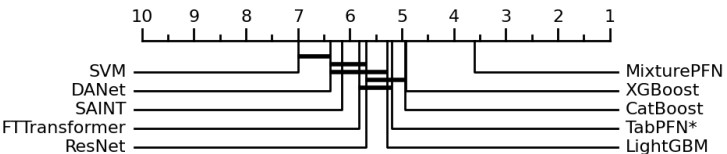

Figure 12: Wilcoxon-Signed Rank Test shows MIXTUREPFN significantly outperforms the Top-10 baselines on the 22 shared datasets, under the accuracy metric. We compute the rank across all 10 cross-validation splits.

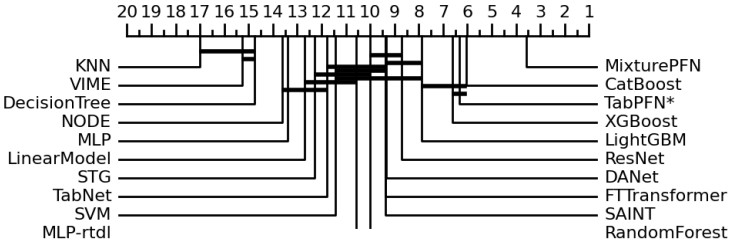

Figure 13: Wilcoxon-Signed Rank Test shows MIXTUREPFN significantly outperforms the all baselines on the 20 shared datasets, under the log likelihood metric. We compute the rank across all 10 cross-validation splits.

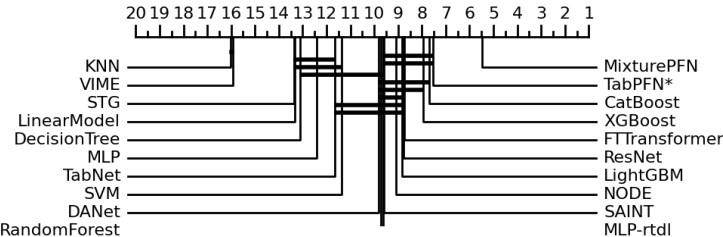

Figure 14: Wilcoxon-Signed Rank Test shows MIXTUREPFN significantly outperforms the all baselines on the 20 shared datasets, under the accuracy metric. We compute the rank across all 10 cross-validation splits.

TUREPFN achieves the best average performance across all datasets. Furthermore, we plot the full range of dataset properties covered, along with the 2 held out datasets from TABZILLA in Figure 16, showing that the datasets successfully run on are representative of the benchmark as a whole. Refer to Section G for more details.

We followed the same experimental setup as TabZilla (McElfresh et al., 2023), which includes: imputing NaN features to its non-NaN mean and all other preprocessing is handled by each respective baseline. MIXTUREPFN and TABPFN* follow TabZilla's PFN preprocessing (McElfresh et al., 2023): categorical features are encoded as ordinal features, outliers are dropped, features are normalized, results are ensembled across shuffling the feature ordering, and results are ensembled across power-law scaled and unscaled features.

## G.2 BASELINE ALGORITHMS

### G.2.1 PRIOR-FITTED NETWORK MODELS (PFN)

TABPFN* is the only PFN-based baseline, which uses a pretrained 12-layer TABPFN transformer model, with embeddings size 512, hidden size 1024 in feed-forward layers, and 4-headed attention. TABPFN is pretrained on a handcrafted dataset prior consisting of randomly generated structural causal models (Hollmann et al., 2022). During inference features and labels are randomly shuffled in batch size 32 then ensembled together, following the TABZILLA benchmark (McElfresh et al., 2023). Recent work extend TabPFN through in-context learning. Most closely related, this MIXTUREPFN was developed in parallel to LoCALPFN (Thomas et al., 2025) which also uses finetuning and retrieval to improve TabPFN's scalability. Unlike LoCALPFN, our work uses different experts for each test-time point, whereas LoCALPFN uses the same expert. Recently, TabDPT (Ma et al., 2024) was proposed to train models to perform TabPFN retrieval. Our work improves TABPFN's scalability and efficacy along different dataset properties, particular number of training samples.

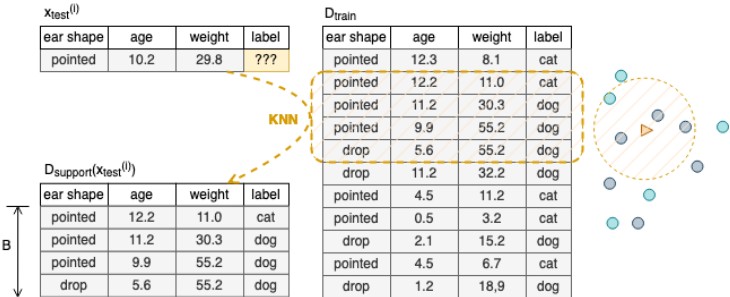

Figure 15: We hypothesize only a subset of the training data, $D_{support}(x_{test}^{(i)})$, is required for effective in-context learning on test sample, $x_{test}^{(i)}$, and this subset is the $B$ nearest training samples in feature space: $D_{support}(x_{test}^{(i)}) = \text{KNN}(x_{test}^{(i)}|D_{train}, B)$.

### G.2.2 GRADIENT-BOOSTED DECISION TREE MODELS (GBDT)

CatBoost (Prokhorenkova et al., 2018), XGBoost (Chen & Guestrin, 2016), and LightGBM (Ke et al., 2017) are GBDT models. These models utilize boosting to construct an ensemble of small trees for evaluation. GBDTs are robust to uninformative or heavy tail features and achieve competitive performance over baselines across different dataset properties. Our work argues in-context learning is a potential competitor against GBDTs, as PFN transformers can potentially learn a better dataset prior than GBDTs.

### G.2.3 DEEP LEARNING ALGORITHMS

ResNet (Gorishniy et al., 2021), MLP-rtdl, TabNet (Arik & Pfister, 2021), MLP, STG (Yamada et al., 2020), VIME (Yoon et al., 2020), NODE (Popov et al., 2019), FTTransformer (Gorishniy et al., 2021), SVM (Cortes & Vapnik, 1995), DANet (Chen et al., 2022) and SAINT (Somepalli et al., 2021) are deep learning-based algorithms. In particular, ResNet is a Convolutional Neural Network designed for tabular learning. MLP-rtdl and MLP are 2 implementations of multilayer-perceptrons. SVM is a support vector machine. TabNet and STG is a neural network architecture that aims to learn GBDT-like operations in a fully differentiable manner. VIME is a gated neural network that is first traing with self supervision. MIXTUREPFN outperforms deep learning algorithms by learning a prior that better regularizes the learning procedure. NODE is a neural network architecture that aims to imitate GBDTs while being fully differentiable and end2end. DANet is a specialized deep learning architecture for tabular data. FTTransformer is a feature encoding transformer model designed for tabular data. SAINT is a self-supervised transformer designed for tabular data.

### G.2.4 SIMPLE ALGORITHMS

RandomForest, DecisionTree, LinearModel, and KNN are all standard machine learning algorithms. We highlight, although MIXTUREPFN is based on evaluating prompts only on KNN neighborhoods, it drastically outperforms KNN. This suggests that In-Context Learning-based models trained on a local neighborhood can outperform both complicated models trained on the entire dataset and simple models that return the average of local neighborhoods. Indeed combining KNN and Large Language Models have been highly successful (Xu et al., 2023; Liu et al., 2021; Guu et al., 2020).

## H ADDITIONAL RESULTS

### H.1 ACCURACY RESULTS AND STANDARD DEVIATIONS

We provide the same Condorcet experiments as the main paper but with the accuracy metric in Table 4. We provide the Condorcet matrix in Figures 7 and 8. These results support the same conclusions found in the main paper.

| Method | Number of Datasets Completed On |
|---|---|
| CatBoost (Prokhorenkova et al., 2018) | 35 |
| XGBoost (Chen & Guestrin, 2016) | 36 |
| MLP-rtdl (Goodfellow et al., 2016; Gorishniy et al., 2021) | 36 |
| MLP (Goodfellow et al., 2016; McElfresh et al., 2023) | 36 |
| ResNet (Gorishniy et al., 2021) | 35 |
| RandomForest (Liaw et al., 2002) | 35 |
| DecisionTree (Quinlan, 1986) | 35 |
| MixturePFN | 34 |
| TabPFN* (Hollmann et al., 2022; McElfresh et al., 2023) | 34 |
| LinearModel (Cox, 1958) | 34 |
| TabNet (Arik & Pfister, 2021) | 33 |
| KNN (Cover & Hart, 1967) | 33 |
| LightGBM (Ke et al., 2017) | 32 |
| VIME (Yoon et al., 2020) | 32 |
| STG (Yamada et al., 2020) | 31 |
| NODE (Popov et al., 2019) | 30 |
| FTTransformer (Gorishniy et al., 2021) | 29 |
| SVM (Cortes & Vapnik, 1995) | 29 |
| SAINT (Somepalli et al., 2021) | 27 |
| DANet (Chen et al., 2022) | 27 |

Table 11: The number of datasets each algorithm completed on across the entire 36 dataset TABZILLA benchmark. Note, the 2 datasets that MIXTUREPFN and TABPFN* (McElfresh et al., 2023) does not run on has too many labels, being unsupported by the pretrained TABPFN (Hollmann et al., 2022). However these 2 datasets are not outliers compared to the 34 datasets that are supported. Note, TABPFN achieves the same results as TABPFN*, except only running on the 17 datasets with ¡3,000 features, hence we compare against the more powerful TABPFN* baseline instead.

|  | Condorcet Wins | Mean Rank | Median Rank | Min Rank | Max Rank |
|---|---|---|---|---|---|
| MIXTUREPFN | 522 | 2.833 | 2 | 1 | 9 |
| TABPFN* | 501 | 4.333 | 4.5 | 2 | 7 |
| TabR | 324 | 17.333 | 18.5 | 8 | 22 |
| FTTransformer | 310 | 6.667 | 6.5 | 3 | 12 |
| MLP-PLR | 223 | 12.417 | 14 | 2 | 19 |

Table 12: Comparison to additional recent deep learning algorithms.

We provide the detailed statistics of the ranking experiments with both log-likelihood and accuracy across all, Top-10, and Top-5 subsets in Tables 5, 6, 7, 8, 9, 10. These results support the same conclusions found in the main paper.

We provide the Wilcoxon-Signed Rank Test with both the log-likelihood and accuracy metric across the all and Top-10 subsets in Figures 12, 13, and 14. These results support the same conclusions found in the main paper.

We provide experiments with even lighter hyperparameter tuning, as discussed in Section J, we call this model, MIXTUREPFN-lite. MIXTUREPFN is tuned over ¿80% less configurations than baselines. MIXTUREPFN-lite is tuned only on 2 hyparparameter settings. We provide the main paper results and Condorcet matrices as presented in Table 18 and Figure 17.

MIXTUREPFN is the Condorcet winner and achieves the top mean rank across all experimental settings, with statistical significance among all and Top-10 subsets. MIXTUREPFN's Log-likelihood results are slightly better, because many algorithms are tied in accuracy across the benchmark. When this occurs, MIXTUREPFN is more confident than baselines when it is correct.

## H.2    ADDITIONAL BASELINES

We implemented and evaluated TabR (Gorishniy et al., 2023) and MLP-PLR (Gorishniy et al., 2022) across 5 hyperparameter settings on the TabZilla benchmark. We show the Condorcet wins and rank evaluated across all algorithms and datasets (now including TabR and MLP-PLR) as reported in Table 1. MIXTUREPFN outperforms recently proposed algorithms under the same hyperparameter tuning budgets. Note, both baselines require heavy hyperparameter tuning in their respective papers. We provide results in Table 12

Compared to LoCALPFN (Thomas et al., 2025), which finetunes a single TabPFN model for retrieval, MIXTUREPFN finetunes a specialized expert on each subset of the training/context data, improving effectiveness. Hence, LoCALPFN is a specific instance of MIXTUREPFN: when $\gamma = \infty$ and there is only one expert. As seen in Table 13, MIXTUREPFN outperforms both TabPFN and LoCALPFN by training a specialized expert on each subset of the training dataset. The more experts that are trained, the better the performance.

we use different types of categorical encodings (target and one-hot encoding), filtering each categorical feature to have at most K dimensions using MRMR, and using different numbers of clusters (where number of experts is proportional to $\gamma$) in Table 14. These additional ablations show performance is more sensitive to the number of experts than choice in categorical encoding algorithms. Hence, it is MIXTUREPFN 's MICP and CaPFN are the core reasons for its superior performance.

We find 3 very large tabular datasets to test our approach. We limit MixturePFN clusters to at most 64 and adopt only target encoding for categorical features. As seen in Table 7, even under these handicaps, MixturePFN consistently improves TabPFN*'s accuracy (dataset size in brackets).

## H.3    NORMALIZED ACCURACY

We evaluate the normalized accuracy across XGBOOST over shared datasets in addition to provided metrics in Table 16, where we find MIXTUREPFN is still the best performing model.

| | Max Experts | $\gamma$ | Electricity Acc | Phoneme Acc. | Airplane Acc. |
|---|---|---|---|---|---|
| TABPFN* | no Finetune | no KNN | 81.2% | 88.3% | 60.0% |
| LoCALPFN† | 1 | $\infty$ | 85.4% | 88.7% | 64.0% |
| MIXTUREPFN † | 1 | 5 | 85.6% | 88.5% | 64.3% |
| MIXTUREPFN † | 8 | 5 | 88.0% | 89.1% | 64.9% |
| MIXTUREPFN | 1024 | 5 | 89.7% | 90.2% | 85.7% |

Table 13: Comparison to recent retrieval models.

| | Encoding-K | $\gamma$ | Electricity Acc | Phoneme Acc. |
|---|---|---|---|---|
| TABPFN* | target-1 | no KNN | 81.2% | 88.3% |
| MIXTUREPFN † | target-1 | 5 | 89.7% | 90.2% |
| MIXTUREPFN † | target-1 | 1 | 87.4% | 87.4% |
| MIXTUREPFN † | onehot-1 | 1 | 86.7% | 86.7% |
| MIXTUREPFN † | target-5 | 1 | 87.4% | 87.8% |
| MIXTUREPFN † | onehot-5 | 1 | 86.3% | 87.9% |

Table 14: Comparison with different categorical encodings.

| | Poker(1MM) | HiggsBig(940K) | BNG(1MM) |
|---|---|---|---|
| MIXTUREPFN | 52.5% | 66.7% | 87.4% |
| TABPFN* | 60.0% | 69.0% | 89.0% |

Table 15: Comparison on additional very large datasets.

| Algorithm | Normalized Accuracy |
|---|---|
| XGBOOST | +0.0% |
| CATBOOST | -0.704% |
| TABPFN* | -3.381% |
| MIXTUREPFN | +0.561% |

Table 16: Normalized Accuracy Results

| Method | Accuracy | K | $\frac{\text{Time}}{\text{Prompt}}$ |
| --- | --- | --- | --- |
| | Mean | Mean | Mean |
| TABPFN* | 83.42% | 1.00 | 1.24s |
| T*+MICP ($\gamma = 1.0$) | 83.96% | 2.25 | 0.90s |
| T*+MICP ($\gamma = 3.0$) | 84.23% | 5.25 | 1.02s |
| T*+MICP ($\gamma = 5.0$) | 84.23% | 8.54 | 0.83s |

Table 17: Trade-off of $\gamma$. $T*$+MICP is short for TABPFN* +MICP. Note as $\gamma$ increases, the accuracy and number of prompters increases, while TABPFN inference time remains constant. Routing costs are negligible with optimized nearest neighbor search (Douze et al., 2024).

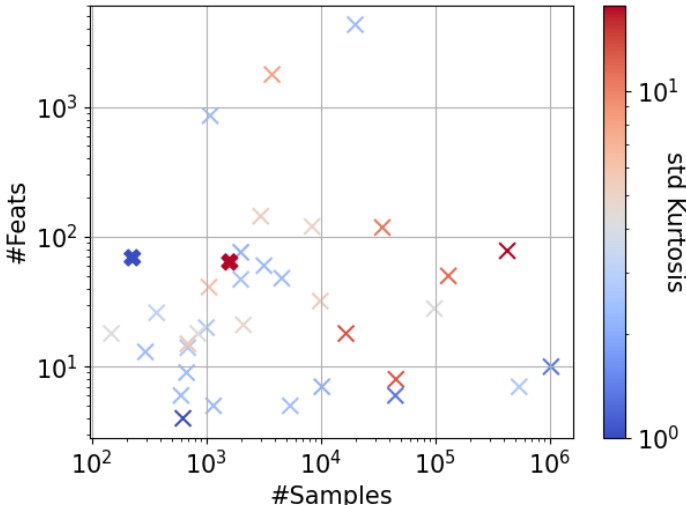

Figure 16: Dataset properties of chosen algorithms from the TABZILLA benchmark. We plot 3 dimensions of the dataset properties of all 36 dataset from TABZILLA. The 2 bold points represent the held-out datasets. As shown, the 34 chosen datasets covers a wide-variety of dataset properties.

## H.4 SENSITIVITY RESULTS ON MORE DATA

We provide the same sensitivity analysis conducted in the main paper but with the accuracy metric in Figure 9. These figures support the same conclusions found in the main paper.

We provide the same sensitivity analysis conducted in the main paper but across the number of features, number of labels, and feature irregularity in Figures 10 and 11. These figures support the same conclusions found in the main paper.

## H.5 EFFICIENCY EFFECTIVENESS TRADE-OFF

$\gamma$ **effectively trade-offs efficiency and effectiveness.** As mentioned in Section 3.1.1, MIXTUREPFN uses a single hyperparameter, $\gamma$, to control the efficiency effectiveness tradeoff. we plot the average accuracy of TABPFN +MICP across $\gamma = [1.0, 3.0, 5.0]$, across the entire dataset. As shown in Table 17, as the hyperparameter $\gamma$ increases, MICP's effectiveness is reliably trade-off for efficiency.

## H.6 DETAILED RESULTS

We provide MIXTUREPFN's and TABPFN*'s accuracies across the 10-folds on all datasets in Tables 20 and 21.

| Method | Condorcet Statistics | | | |
|---|---|---|---|---|
| | #Votes↑ | #Wins↑ | #Ties | #Losses↓ |
| MixturePFN-lite | **503** | **19** | 0 | **0** |
| XGBoost | 502 | 18 | 0 | 1 |
| CatBoost | 479 | 17 | 0 | 2 |
| SAINT | 404 | 16 | 0 | 3 |
| TabPFN* | 385 | 13 | 1 | 5 |
| LightGBM | 374 | 14 | 1 | 4 |
| DANet | 312 | 14 | 0 | 5 |
| FTTransformer | 295 | 12 | 0 | 7 |
| ResNet | 287 | 10 | 0 | 9 |
| SVM | 286 | 11 | 0 | 8 |
| STG | 286 | 9 | 0 | 10 |
| RandomForest | 248 | 7 | 0 | 12 |
| NODE | 244 | 7 | 0 | 12 |
| MLP-rtdl | 228 | 5 | 0 | 14 |
| TabNet | 210 | 5 | 0 | 14 |
| LinearModel | 202 | 3 | 1 | 15 |
| MLP | 193 | 5 | 1 | 13 |
| VIME | 134 | 2 | 0 | 17 |
| DecisionTree | 115 | 1 | 0 | 18 |
| KNN | 74 | 0 | 0 | 19 |

Table 18: MIXTUREPFN is the Condorcet winner across 36 datasets against 19 baseline algorithms. We rank algorithms based on their log-likelihoods.

## I   IMPLEMENTATION

We implemented MICP by first preprocessing the train data into separate prompts via KNN, chunking each prompt into batches, then called TabZilla APIs to run the desired PFN model on each batch. We implemented CAPFN by bootstrapping our training dataset then running maximum likelihood loss on the bootstrapped datasets. MIXTUREPFN's implementation is built off the official TABZILLA codebase (git, 2024b) and will be open-sourced on Github.

### I.1   OPTIMIZING TABPFN'S IMPLEMENTATION

TABZILLA only obtained TABPFN* results on 7 out of 34 benchmark datasets (McElfresh et al., 2023), due to memory constraints. We identified an implementation inefficiency where "prompts" are constructed with the entire test dataset, i.e. $(X_{test}, D_{train})$, causing memory overflow. We optimized TABPFN*'s implementation by batching test samples, $(X_{batch}|D_{train}), X_{batch} \subseteq X_{test}$, with batch size 1024, and report results over all 26 datasets.

## J   HYPERPARAMETER SETUP

As TABPFN transformers can handle up to 3,000 training samples, we set $B = 3,000$. We empirically found the minimum number of iterations and batch-size required for loss convergence on the artificial-characters dataset to be 128 iterations and $N_{batch} = 64$, which we set for all other datasets. During inference, we use a larger batch size, $N_{batch} = 1024$, as gradients no longer need to be stored. We finetuned the model using the Adam optimizer with a learning rate of 0.001. As TABPFN transformers can handle up to 100 features, for datasets with over 100 features and TABPFN-based models, we use Maximum Relevance and Minimum Redundancy (mRMR) feature selection (Ding & Peng, 2005) to reduce the number of features to 100. We follow the TABZILLA benchmark, setting $N_{ensemble} = 16$, which shuffles features $N_{ensemble}/2$ times for both the original and applies power-law scaled features. MIXTUREPFN's router was implemented using the FAISS (Douze et al., 2024) library.

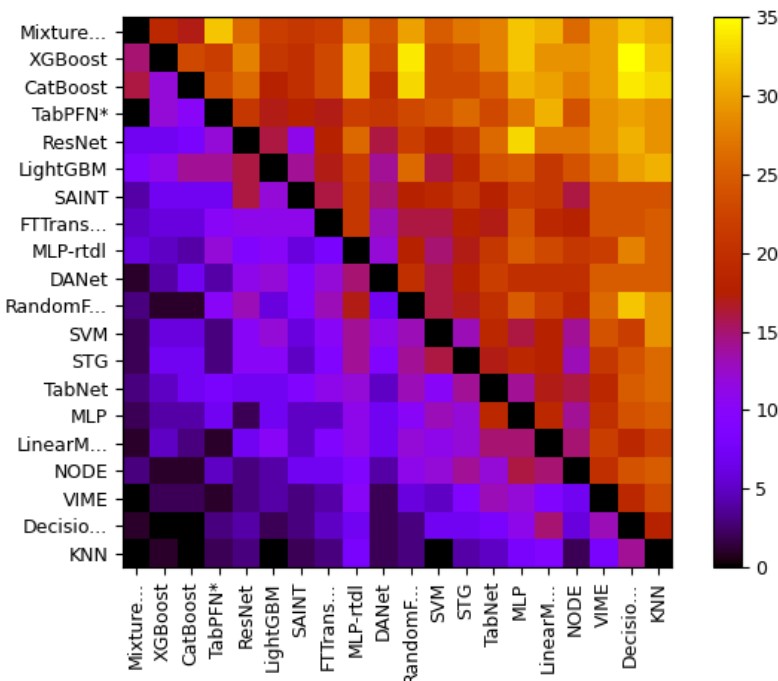

Figure 17: Pairwise comparison matrix for Condorcet voting over the log likelihood metric with lightly tuned MIXTUREPFN. Note, MIXTUREPFN-lite is the Condorcet winner.

Due to the large variability in datasets in the TABZILLA benchmark, we try 4 hyperparameter settings: (1) $\gamma = 5.0$, (2) $\gamma = 1.0$, (3) $\gamma = 1.0$ but MRMR with 50 features instead of 100 features for feature count scalability, and (4) $\gamma = 1.0$ but with Catboost instead of Ordinal encoding for categorical feature scalability (Hollmann et al., 2022). Hyperparameters are chosen by picking the setting which maximizes performance on the validation set. Models are evaluated on the test set, which is not seen during hyperparameter tuning. In contrast to all other baselines, which are tuned across 30 hyperparameter settings, MIXTUREPFN performs **much less** hyperparmeter tuning than baselines. Baseline hyperparameter settings are the same as the TABZILLA (McElfresh et al., 2023) benchmark. Note, even if only the $\gamma$ parameter tuned (i.e. only settings (1) and (2)), MIXTUREPFN is still much better than TABPFN, as presented in Table 18 and Figure 17. All results were collected over 10-folds following TABZILLA (McElfresh et al., 2023) and OpenML. We tune the hyperparameters by splitting the train set of each fold into training and validation following TABZILLA (McElfresh et al., 2023). Ablation studies were performed modifying hyperparameter setting (1). Dataset preprocessing details can be found in Appendix G.1.

## K HARDWARE

All experiments were conducted on an Nvidia V100 GPU and an AMD EPYC 7402 CPU. Each experiment is given a budget of 10 hours for a single dataset and algorithm.

| Dataset | Dataset Properties | | | | Top 2 Algs. | |
|---|---|---|---|---|---|---|
| | #Samples | #Feats | #Lab. | Std. Kurt. | 1st | 2nd |
| lymph | 148 | 18 | 4 | 17.04 | XGBoost | TABPFN* |
| audiology | 226 | 69 | 24 | None | XGBoost | - |
| heart-h | 294 | 13 | 1 | None | MLP-rtdl | MIXTUREPFN |
| colic | 368 | 26 | 1 | 4.0 | XGBoost | MIXTUREPFN |
| monks-prob... | 601 | 6 | 1 | None | MIXTUREPFN | MLP-rtdl |
| balance-scale | 625 | 4 | 3 | 0.02 | MIXTUREPFN | TABPFN* |
| profb | 672 | 9 | 1 | 0.95 | MIXTUREPFN | MLP-rtdl |
| Australian | 690 | 14 | 1 | 2.0 | XGBoost | TABPFN* |
| credit-approval | 690 | 15 | 1 | 74.77 | TABPFN* | MIXTUREPFN |
| vehicle | 846 | 18 | 4 | 15.16 | MIXTUREPFN | TABPFN* |
| credit-g | 1000 | 20 | 1 | 1.92 | MIXTUREPFN | TABPFN* |
| qsar-biodeg | 1055 | 41 | 1 | 93.24 | MIXTUREPFN | TABPFN* |
| cnae-9 | 1080 | 856 | 9 | None | MLP-rtdl | MLP |
| socmob | 1156 | 5 | 1 | None | XGBoost | TABPFN* |
| 100plants | 1599 | 64 | 100 | 17.66 | XGBoost | - |
| mfeat-fourier | 2000 | 76 | 10 | 0.64 | MIXTUREPFN | TABPFN* |
| mfeat-zernike | 2000 | 47 | 10 | 1.42 | MIXTUREPFN | TABPFN* |
| kc1 | 2109 | 21 | 1 | 28.34 | MIXTUREPFN | TABPFN* |
| jasmine | 2984 | 144 | 1 | 47.6 | MIXTUREPFN | XGBoost |
| splice | 3190 | 60 | 3 | None | MIXTUREPFN | XGBoost |
| Bioresponse | 3751 | 1776 | 1 | 328.77 | XGBoost | MIXTUREPFN |
| ada-agnostic | 4562 | 48 | 1 | None | XGBoost | MIXTUREPFN |
| phoneme | 5404 | 5 | 1 | 1.23 | MIXTUREPFN | XGBoost |
| SpeedDating | 8378 | 120 | 1 | 36.43 | MIXTUREPFN | XGBoost |
| GesturePhase... | 9873 | 32 | 5 | 52.18 | MIXTUREPFN | XGBoost |
| artificial-char... | 10218 | 7 | 10 | 0.63 | XGBoost | MIXTUREPFN |
| elevators | 16599 | 18 | 1 | 2986.5 | MIXTUREPFN | TABPFN* |
| guillermo | 20000 | 4296 | 1 | None | XGBoost | TABPFN* |
| nomao | 34465 | 118 | 1 | 1100.34 | XGBoost | MIXTUREPFN |
| jungle-chess... | 44819 | 6 | 3 | 0.08 | MIXTUREPFN | XGBoost |
| electricity | 45312 | 8 | 1 | 2693.51 | XGBoost | MIXTUREPFN |
| higgs | 98050 | 28 | 1 | 15.53 | XGBoost | MLP |
| MiniBooNE | 130064 | 50 | 1 | 1686.9 | MIXTUREPFN | XGBoost |
| albert | 425240 | 78 | 1 | 12162.65 | MIXTUREPFN | XGBoost |
| airlines | 539383 | 7 | 1 | 2.01 | MIXTUREPFN | XGBoost |
| poker-hand | 1025009 | 10 | 10 | 0.08 | XGBoost | MIXTUREPFN |

Table 19: Dataset statistics for valid TABZILLA benchmark datasets. Ranks are computed across algorithms that run on all datasets MIXTUREPFN runs on: MIXTUREPFN, TABPFN*, XGBOOST, MLP, and MLP-rtdl. Note this list of datasets was originally curated from 197 datasets, to contain only those difficult for all models. We list the top 2 performing algorithms based on log likelihood, following TABZILLA, on each dataset. MIXTUREPFN achieves state-of-the-art performance.

| Dataset | Model | Mean ± Std Accuracy ↑ |
|---------|-------|------------------------|
| australian | TabPFN* | 0.868±0.036 |
| | MixturePFN | 0.861±0.023 |
| bioresponse | TabPFN* | 0.791±0.018 |
| | MixturePFN | 0.793±0.017 |
| gesturepha... | TabPFN* | 0.569±0.014 |
| | MixturePFN | 0.704±0.011 |
| miniboone | TabPFN* | 0.927±0.003 |
| | MixturePFN | 0.946±0.003 |
| speeddating | TabPFN* | 0.856±0.006 |
| | MixturePFN | 0.887±0.011 |
| ada-agnostic | TabPFN* | 0.845±0.016 |
| | MixturePFN | 0.842±0.013 |
| airlines | TabPFN* | 0.600±0.003 |
| | MixturePFN | 0.857±0.005 |
| albert | TabPFN* | 0.638±0.005 |
| | MixturePFN | 0.903±0.003 |
| artificial... | TabPFN* | 0.650±0.013 |
| | MixturePFN | 0.717±0.008 |
| balance-scale | TabPFN* | 0.989±0.013 |
| | MixturePFN | 0.997±0.010 |
| cnae-9 | TabPFN* | 0.896±0.029 |
| | MixturePFN | 0.899±0.029 |
| colic | TabPFN* | 0.823±0.044 |
| | MixturePFN | 0.807±0.067 |
| credit-app... | TabPFN* | 0.884±0.050 |
| | MixturePFN | 0.872±0.053 |
| credit-g | TabPFN* | 0.729±0.028 |
| | MixturePFN | 0.740±0.021 |
| electricity | TabPFN* | 0.812±0.005 |
| | MixturePFN | 0.897±0.003 |
| elevators | TabPFN* | 0.900±0.006 |
| | MixturePFN | 0.905±0.005 |
| guillermo | TabPFN* | 0.791±0.013 |
| | MixturePFN | 0.799±0.018 |
| heart-h | TabPFN* | 0.837±0.044 |
| | MixturePFN | 0.834±0.046 |
| higgs | TabPFN* | 0.665±0.007 |
| | MixturePFN | 0.693±0.005 |
| jasmine | TabPFN* | 0.804±0.016 |
| | MixturePFN | 0.861±0.008 |
| jungle-che... | TabPFN* | 0.823±0.006 |
| | MixturePFN | 0.865±0.004 |
| kc1 | TabPFN* | 0.862±0.011 |
| | MixturePFN | 0.866±0.013 |
| lymph | TabPFN* | 0.810±0.096 |
| | MixturePFN | 0.810±0.096 |
| mfeat-fourier | TabPFN* | 0.828±0.025 |
| | MixturePFN | 0.847±0.025 |

Table 20: Mean and Std. Accuracy of MIXTUREPFN and TABPFN* on all datasets across 10-folds (part 1).

| Dataset | Model | Mean $\pm$ Std Accuracy $\uparrow$ |
|---|---|---|
| mfeat-zernike | TabPFN* | 0.828±0.015 |
| | MixturePFN | 0.846±0.024 |
| monks-prob... | TabPFN* | 1.000±0.000 |
| | MixturePFN | 1.000±0.000 |
| nomao | TabPFN* | 0.953±0.003 |
| | MixturePFN | 0.966±0.002 |
| phoneme | TabPFN* | 0.883±0.014 |
| | MixturePFN | 0.902±0.015 |
| poker-hand | TabPFN* | 0.517±0.011 |
| | MixturePFN | 0.677±0.002 |
| profb | TabPFN* | 0.691±0.028 |
| | MixturePFN | 0.685±0.024 |
| qsar-biodeg | TabPFN* | 0.885±0.033 |
| | MixturePFN | 0.883±0.038 |
| socmob | TabPFN* | 0.933±0.016 |
| | MixturePFN | 0.929±0.017 |
| splice | TabPFN* | 0.876±0.018 |
| | MixturePFN | 0.983±0.005 |
| vehicle | TabPFN* | 0.847±0.023 |
| | MixturePFN | 0.847±0.024 |

Table 21: Mean and Std. Accuracy of MIXTUREPFN and TABPFN* on all datasets across 10-folds (part 2).

