# OpenReview forum: "Mixture of In-Context Prompters for Tabular PFNs"
_ICLR.cc/2025/Conference — ICLR 2025 Poster_

### Official Review · Reviewer_X42j · 2024-10-28

**Soundness:** 3
**Presentation:** 3
**Contribution:** 3
**Rating:** 6
**Confidence:** 3

**Summary:**

In this paper, the authors propose MixturePFN, an extension of Sparse Mixture of Experts to TabPFN to alleviate the context size limitations of the existing TabPFN. On the TabZilla benchmark, MixturePFN outperforms state-of-the-art tabular prediction models.

**Strengths:**

1. The idea of Mixture of Experts blending into TabPFN seems novel.

2. The effectiveness of MixturePFN is well evaluated in well-established benchmarks against a variety of baseline methods.

3. Writing is easy to follow.

**Weaknesses:**

1. The biggest weakness I think is that the paper is missing a comparison with LoCalPFN [1]. Since LoCalPFN also tries to make TabPFN effective even on datasets with many-shots, I think it should be mentioned in the paper.

----
[1] Thomas et al., Retrieval & Fine-Tuning for In-Context Tabular Models, NeurIPS 2024

**Questions:**

1. Can you provide a comparison with LoCalPFN [1]? If not possible, I think the comparison should be done using k neighbor samples rather than random sampling, at least for TabPFN*.

2. I see that the authors say in the limitations section that they didn't do it on a dataset with a million samples, but I'm somewhat curious about the effectiveness of MixturePFN on a dataset with a million samples, since the paper is aimed at the scale-up aspect.

3. I'm also curious about the effectiveness of MixturePFN on datasets with hundreds or thousands of features, which is very practical in the real world.

----
[1] Thomas et al., Retrieval & Fine-Tuning for In-Context Tabular Models, NeurIPS 2024

---

> ### Author Response · Authors · 2024-11-21
> **Thank you for your review!**
>
> W1 + Q1. Regarding LoCALPFN, please refer to the **[LoCALPFN is a parallel work]** and
> **[Comparison with LoCALPFN]** common questions.
>
> We highlight: (1) LoCALPFN’s NeurIPS paper **was accepted after** the ICLR submission deadline, (2) MixturePFN was **put on arXiv before** LoCALPFN, and (3) after discussing with the authors of LoCALPFN, **the authors of LoCALPFN agree with us** that MixturePFN and LoCALPFN came to this conclusion contemporaneously.
>
> As requested, we provide empirical comparison with LoCALPFN (i.e. KNN prompting + finetuning). We show that by using more experts, MixturePFN substantially improves LoCALPFN’s performance. Our experimental results are shown in the **[Comparison with LoCALPFN]** common question.
>
> Q2. We find 3 very large tabular datasets to test our approach. Due to the limited time of the rebuttal, we limit MixturePFN clusters to at most 64 and adopt only target encoding for categorical features. Even under these handicaps, MixturePFN consistently improves TabPFN*’s accuracy (dataset size in brackets):
>
>
> ```
> Model Name   | Poker(1MM) | HiggsBig(940K) | BNG(1MM)
> -------------+------------+----------------+----------
> TabPFN*      | 52.5%      | 66.7%          | 87.4%
> MixturePFN   | 60.0%      | 69.0%          | 89.0%
> ```
>
> Q3. As shown in Figures 10c and 11c of the Appendix, we test on datasets with hundreds or thousands of features. We find MixturePFN and deep learning perform worse on datasets with more features. In theory, as long as (1) most features are noise and (2) MRMR successfully filters noise from useful features, performance should not deteriorate. Our experiments (Figures 10c and 11c of the Appendix) suggest either (1) or (2) is violated on these datasets. Thus we recommend improved feature selection or feature compression algorithms to extend our work to datasets with many features for practitioners
>
> Thank you again for your thorough review. Considering our rebuttal, we would highly appreciate your raising scores and further support for our paper!

---

> > ### Comment · Reviewer_X42j · 2024-11-22
> > **Thank you for your response**
> >
> > Thank you for putting a lot of time and effort into comparing LoCalPFN to your work, MixturePFN. I agree that your work is contemporary to LoCalPFN, but I am still impressed that MixturePFN surpasses that approach. I encourage the authors to include the results in a revised manuscript, and I give an acceptance grade on this point.

---

> > > ### Author Response · Authors · 2024-11-22
> > > **Thank you!**
> > >
> > > Thank you for raising your score! We are glad our rebuttal addressed your concerns, and will include these results in our revised manuscript.

---

### Official Review · Reviewer_DZw6 · 2024-11-04

**Soundness:** 3
**Presentation:** 3
**Contribution:** 3
**Rating:** 8
**Confidence:** 2

**Summary:**

The paper proposes the MixturePFN framework, which extends TabPFN for large tabular datasets by addressing the performance and scalability limitations of the number of table rows. The authors propose:
1. Mixture of In-Context Prompters (MICP), which optimizes inference by using a sparse mixture of experts to route test samples to specific "prompters" that create context-specific prompts to separate large training datasets into manageable clusters.
2. Context-Aware Finetuning (CAPFN), which addresses distributional shift issues by specializing each prompter on its assigned
context via parameter efficient finetuning.

**Strengths:**

- The MICP strategy effectively reduces memory usage, allowing the model to handle larger datasets compared to existing TabPFN
- CAPFN bootstrapping and finetuning approach appears to be an effective way to mitigate distribution shift ICL for tabular data
- Extensive benchmarks against 19 strong baselines show good performance in both mean rank and Condorcet ranking across diverse datasets

**Weaknesses:**

- While MIXTUREPFN improves dataset scalability, it still struggles with feature-rich datasets, potentially limiting its applicability in domains with high-dimensional data, such as patient healthcare data. I realize the authors leave this to future work, but this is an area where simple XGBoost performs quite well, and I would be curious about their thoughts on tackling this issue.

- MICP's reliance on K-Means clustering to segment data into meaningful clusters as the quality of clusters can vary significantly based on dataset properties / distance metric chosen. Poor clustering could lead to suboptimal routing and ineffective prompts for certain test samples. I'd be curious to see some ablations in this area.

- The CAPFN bootstrapping method might introduce biases or overfitting if the sampled subsets are not representative of the entire dataset. Bootstrapping from small clusters may fail to capture enough diversity, especially in cases with imbalanced classes or rare features. I'd be also curious to see how this method works with highly imbalanced labels e.g. 1\% positive.

**Questions:**

See weaknesses.

Can categorical features simply be encoded as ordinal features? Is that not implying false relationships between unordered elements?

---

> ### Author Response · Authors · 2024-11-21
> **Thank you for your review!**
>
> W1. Yes, feature scalability is an inherent limitation of TabPFN. We believe there are several future directions: (1) improved feature selection algorithms improving MRMR, (2) improved categorical encoding algorithms to reduce cardinality of large categorical features, and (3) efficiently addressing TabPFN’s underlying feature invariance problem with architectural changes. Because these directions are outside the scope of our paper, which studies Mixture of Experts on TabPFN, we leave these directions to future work.
>
> W2. We agree that ideal clustering algorithms are a related but orthogonal problem to our work. As requested, we perform several ablation studies to test this: we use different types of categorical encodings (target and one-hot encoding), filtering each categorical feature to have at most K dimensions using MRMR, and using different numbers of clusters (where number of experts is proportional to gamma).
>
> ```
> Model Name  | encoding-K | gamma  | elec. Acc | phon. Acc
> ------------+------------+--------+-----------+-----------
> TabPFN*     | target-1   | no KNN | 81.2%     | 88.3%
> MixturePFN  | target-1   | 5      | 89.7%     | 90.2%
> MixturePFN  | target-1   | 1      | 87.4%     | 87.4%
> MixturePFN  | onehot-1   | 1      | 86.7%     | 86.7%
> MixturePFN  | target-5   | 1      | 87.4%     | 87.8%
> MixturePFN  | onehot-5   | 1      | 86.3%     | 87.9%
> ```
>
> These additional ablations show performance is more sensitive to the number of experts than choice in categorical encoding algorithms. Hence, it is MixturePFN’s core novelties (MICP and CaPFN) are the reasons for its superior performance.
>
> W3. We agree with your intuition! When we plot the model performance improvement over TabPFN w.r.t. class imbalance, we find a loose correlation (R2=0.249). Hence, our method performs better on balanced data. We will add this interesting finding to our paper!
>
> Thank you again for your detailed review! If our rebuttal answers your concerns, we highly appreciate your support for our paper.

---

> ### Comment · Reviewer_DZw6 · 2024-11-26
>
> Thank you for the additional experiments, I also emphasize with the contemporaneous submission with LoCALPFN, and am impressed that results were able to be run in a short amount of time. I have updated my score accordingly. Best of luck!

---

> > ### Author Response · Authors · 2024-11-26
> > **Thank you!**
> >
> > Thank you for raising your score! We are glad that our additional results and related work comparison help address your concerns and will include your valuable suggestions in our revised manuscript.

---

### Official Review · Reviewer_5Xhm · 2024-11-04

**Soundness:** 3
**Presentation:** 3
**Contribution:** 1
**Rating:** 6
**Confidence:** 2

**Summary:**

The paper proposes a mixture of experts approach for in-context learning on tabular data. Each expert in the mixture is a K-means cluster and the model routes the input instance to the closest cluster. This addresses the problem of context size in large datasets and provides a better selection of prompt instance than random sampling. To adapt the model to this type of routing authors also propose fine tuning by selecting a cluster of each training instance and maximizing the likelihood.

**Strengths:**

The paper is well written and proposes a justified solution to address the context length issue for in-context learning models such as TabPFN. Authors conduct extensive experiments on many real world dataset to demonstrate the effectiveness of the proposed approach and compare with leading tree-based and deep learning tabular methods.

**Weaknesses:**

There is a very related previous work "Retrieval & Fine-Tuning for In-Context Tabular Models" by Thomas et al, which proposes both nearest neighbor retrieval to improve the prompt and fine tuning with this approach to adapt the model to the target distribution. I think the authors have to compare with this work and highlight what is novel in MixturePFN.

**Questions:**

I could not find an ablation study on the number of clusters K vs model performance, have you done these experiments?

---

> ### Author Response · Authors · 2024-11-21
> **Thank you for your review!**
>
> W1. Regarding “Retrieval & Fine-tuning for In-Context Tabular Models” (i.e. LoCALPFN), please refer to the common response **[LoCALPFN is a parallel work]**.
>
> According to ICLR policy, LoCALPFN is a parallel work to our work, MixturePFN. We highlight: (1) LoCALPFN’s NeurIPS paper **was accepted after** the ICLR submission deadline, (2) MixturePFN (which proposes *“nearest neighbor retrieval to improve the prompt and fine tuning”*) was **put on arXiv before** LoCALPFN, and (3) after discussing with the authors of LoCALPFN, **the authors of LoCALPFN agree with us** that MixturePFN and LoCALPFN came to this conclusion contemporaneously.
>
> Nonetheless, we highlight the difference: compared to LoCALPFN, which finetunes a single TabPFN model for retrieval, MixturePFN finetunes a specialized expert on each subset of the training/context data, improving effectiveness. We find this difference substantially improves tabular classification accuracy. We provide empirical results in the common response **[Comparison with LoCALPFN]**.
>
> Q1. Yes, we provide ablation studies on the number of experts vs model performance in Table 14 of the Appendix, which computes average accuracy across shared datasets with different number of experts (where the number of clusters = gamma * dataset_size). Our experiments show as the number of experts increases, performance improves.
>
> ```
> gamma | accuracy
> ------+----------
> 0.0   | 83.42%
> 1.0   | 83.96%
> 3.0   | 84.23%
> 5.0   | 84.23%
> ```
>
> This trend is also supported by our new ablation studies in the **[Comparison with LoCALPFN]** common question, where we limit the total number of experts (i.e. total number of clusters stays constant, but each expert is responsible for multiple clusters).
>
> We greatly appreciate your efforts and expertise on tabular in-context learning and hope our explanation of differences and detailed timeline between MixturePFN and LoCALPFN demonstrate why MixturePFN is novel. If you agree, we would greatly appreciate raising your score! Thank you!

---

> > ### Author Response · Authors · 2024-11-27
> > **Following-Up**
> >
> > Thank you once again for your thorough review and literature search. We will incorporate your valuable suggestions into our work and look forward to hearing your feedback!

---

> > > ### Author Response · Authors · 2024-12-02
> > > **Following-Up**
> > >
> > > Thank you once again for your thorough review and literature search! As stated in our original rebuttal, [1] is a contemporaneous work. Nonetheless, MixturePFN's design choices significantly improves [1] performance. As the end of the Author-Reviewer discussion period nears, we look forward to receiving your feedback on our response!
> > >
> > > [1] Retrieval & Fine-tuning for In-Context Tabular Models

---

> > > > ### Comment · Reviewer_5Xhm · 2024-12-03
> > > > **Thanks**
> > > >
> > > > Thanks for a detailed response and comparison with LoCALPFN, I agree that the two works are contemporaneous and have revised my score.

---

> > > > > ### Author Response · Authors · 2024-12-03
> > > > > **Thank you!**
> > > > >
> > > > > Thank you for raising your score! We will include these comparisons in our work, following your helpful suggestions.

---

### Author Response · Authors · 2024-11-21
**Common Response**

**[LoCALPFN is a parallel work]**

**Please Read:** Thank you very much for your thorough efforts reviewing our paper and its related works! According to ICLR policy, MixturePFN and LoCALPFN are contemporaneous: *“We consider papers contemporaneous if they are published within the last four months. That means, since our full paper deadline is October 1, if a paper was published (i.e., at a peer-reviewed venue) on or after July 1, 2024, authors are not required to compare their own work to that paper.”*.

In fact, MixturePFN was the first to post on arXiv. Here is a detailed timeline:


- MixturePFN was posted on arXiv in May 2024, proposing MoE KNN and finetuning.

- LoCALPFN was posted on arXiv in June 2024, proposing KNN retrieval and finetuning.

- MixturePFN was submitted to ICLR on October 1 2024.

- LoCALPFN (camera-ready) was accepted to NeurIPS on October 30 2024. After discussing with the authors of LoCALPFN, we both agree our papers are contemporaneous.

We hope our timeline gives a better perspective on the relationship between MixturePFN and LoCALPFN. Although we are *not required to*, we still provide a design and empirical comparison with LoCALPFN for completeness in the next section.

**[Comparison with LoCALPFN]**

Compared to LoCALPFN, which finetunes a single TabPFN model for retrieval, MixturePFN finetunes a specialized expert on each subset of the training/context data, improving effectiveness. Hence, LoCALPFN is a specific instance of MixturePFN: when gamma=inf and there is only one expert.

Because LoCALPFN does not release source code or performance on individual datasets (at the time of this rebuttal), we reimplement LoCALPFN† under our framework by setting gamma=inf (i.e. KNN retrieval) and limiting the number of experts. We compute average accuracy across several datasets: electricity, phoneme, and airlines.

```
Model Name  | max experts | gamma  | elec. Acc | phon. Acc | air. Acc
------------+-------------+--------+-----------+-----------+----------
TabPFN*     | no finetune | no KNN | 81.2%     | 88.3%     | 60.0%
LoCALPFN†   | 1           | inf    | 85.4%     | 88.7%     | 64.0%
MixturePFN† | 1           | 5      | 85.6%     | 88.5%     | 64.3%
MixturePFN† | 8           | 5      | 88.0%     | 89.1%     | 64.9%
MixturePFN  | 1024        | 5      | 89.7%     | 90.2%     | 85.7%
```

As seen above, MixturePFN outperforms both TabPFN and LoCALPFN by training a specialized expert on each subset of the training dataset. The more experts that are trained, the better the performance.

---

### Comment · Area_Chair_J1ho · 2024-11-24
**Reminder: Author-Reviewer Discussion Period Closing Soon**

This is a reminder that the author-reviewer discussion period will end on Nov 26 AoE.

Your engagement during this phase is critical for providing valuable feedback and clarifications. If you have any remaining questions or comments, please take a moment to participate before the deadline.

Thank you for your contributions to this important process.

AC

---

### Meta-Review · Area_Chair_J1ho · 2024-12-21

**Metareview:**

(a) Summary of Scientific Claims and Findings

The paper presents MixturePFN, an enhancement of Sparse Mixture of Experts tailored for tabular Prior-Fitted Networks (TabPFNs). MixturePFN leverages specialized In-Context Learning (ICL) experts, applied to clusters of tabular data, to improve both efficiency and accuracy, particularly on large datasets. The framework consists of two key components: Mixture of In-Context Prompters (MICP): Routes test samples to appropriate clusters, enabling effective segmentation of large datasets, and Context-Aware Fine-Tuning (CAPFN): Mitigates distributional shifts by employing fine-tuning specific to each cluster.

(b) Strengths of the Paper

1. Innovative integration of Mixture of Experts with ICL, designed specifically for tabular data applications.

2. Demonstrates substantial performance improvements over state-of-the-art models, supported by extensive validation on diverse and large datasets.

(c) Weaknesses of the Paper and Missing Elements

1. Limited scalability to datasets with high-dimensional features, such as those found in healthcare.

2. CAPFN may introduce biases due to insufficient cluster diversity, which can be problematic for datasets with imbalanced classes.

(d) Decision and Rationale

The paper offers a significant and well-substantiated contribution to advancing ICL for tabular data. While there are concerns about scalability and generalization, the paper’s strengths and innovative approach outweigh these limitations.

**Additional Comments On Reviewer Discussion:**

Consensus on the originality and impact of MixturePFN, with reviewers recognizing its contemporaneity with LoCALPFN as a parallel work.

Recommendations for addressing scalability issues in feature-rich datasets and improving the robustness of clustering techniques.

---

### Decision · Program_Chairs · 2025-01-22

Accept (Poster)